# Directional Bias Helps Stochastic Gradient Descent to Generalize in Nonparametric Model

## Abstract

This paper studies the Stochastic Gradient Descent (SGD) algorithm in kernel regression. The main finding is that SGD with moderate and annealing step size converges in the direction of the eigenvector that corresponds to the largest eigenvalue of the gram matrix. On the contrary, the Gradient Descent (GD) with a moderate or small step size converges along the direction that corresponds to the smallest eigenvalue. For a general squared risk minimization problem, we show that directional bias towards a larger eigenvalue of the Hessian (which is the gram matrix in our case) results in an estimator that is closer to the ground truth. Adopting this result to kernel regression, the directional bias helps the SGD estimator generalize better. This result gives one way to explain how noise helps in generalization when learning with a nontrivial step size, which may be useful for promoting further understanding of stochastic algorithms in deep learning. The correctness of our theory is supported by simulations and experiments of Neural Network on the FashionMNIST dataset.

## 1 Introduction

In this paper, we study the Stochastic Gradient Descent (SGD) algorithm in a nonparametric regression model. Among nonparametric models, one popular choice in both statistics and machine learning communities is the kernel model that is generated by a Reproducing Kernel Hilbert Space (RKHS). When fitting kernel models to the data, SGD is computationally efficient as compared to Gradient Descent (GD) (Ma et al., 2018). This motivates us to analyze the properties of SGD in the kernel model, especially for SGD with a nontrivial step size for a practical reason. In particular, we aim to provide a fundamental explanation of why SGD estimators generalize well.

Our work is inspired by Wu et al. (2021), which shows that directional bias has a significant impact on the generalization property in the linear regression model. We adopt a similar concept of the directional bias, but generalize them to the nonparametric model. Our result is a non-trivial extension of their approach, due to the difference in our problem setting and our SGD algorithm design. To the best of our knowledge, we are the first to show the directional bias phenomenon of SGD and analyze how it helps generalization in nonparametric regression.

**Our contributions** are two folded. First, we study the directional bias of (S)GD in a nonparametric regression model. On the one hand, nonparametric regression is well studied in both statistics and machine learning. On the other hand, the directional bias is a relatively new concept (Wu et al., 2021) of an algorithm that affects the statistics properties, and there is no thorough understanding of the directional bias of (S)GD algorithms for the nonparametric regression model. Note that our result is closely related to those in Belkin et al. (2018); Liang & Rakhlin (2020): they prove that SGD and GD algorithms both converge to the minimum norm interpolate, where the same properties are discovered for SGD and GD; whereas we investigate the solution paths before their convergence, and show that SGD and GD have different solution paths that lead to different properties. Our result helps to explain why the SGD generalizes better than GD.

Our second contribution is to unify the conditions to show the directional bias of (S)GD sequences in nonparametric models. The main condition is the diagonal dominant gram matrix, which covers a large class of kernel functions and allows us to study their properties. Moreover, our SGD is

different from those in Wu et al. (2021): they define SGD in epochs while we define SGD in steps. The fundamental difference in the SGD algorithm requires us to develop different techniques for analyzing the SGD sequence and showing its directional bias, which has not yet been covered in the current literature.

**Main Theorems** of this paper can be divided into two parts, briefly summarized as follows:

**First** is the directional bias of SGD. Theorem 5 shows that for a two-stage SGD with a moderate step size in the first stage and a small step size in the second stage, an early-stopped estimator has a directional bias towards the eigenvector that corresponds to the largest eigenvalue of the gram matrix. Later we refer to this direction as the direction towards the largest eigenvector to simplify the statement.

As a comparison, Theorem 7 shows that GD with both moderate and small step sizes has a directional bias towards the eigenvector that corresponds to the smallest eigenvalue of the gram matrix (denote it as the direction towards the smallest eigenvector). From which, we conclude that SGD and GD have different direction biases in kernel regression.

**Second** is the implication of directional bias. The implication is very useful since it quantifies the effect of the directional bias on the generalization error. Theorem 9 considers a general problem of quadratic loss. It shows that the estimator biased towards the largest eigenvector of the Hessian (which is the gram matrix in our nonparametric regression) can have the smallest parameter estimation error, when compared with other estimators of the same loss.

With this high-level idea of directional bias helps generalization, Theorem 11 compares the generalization error of SGD and GD in our problem setting. In particular, it upper bounds the generalization error of SGD and lower bounds the generalization error of GD. By directly comparing the error bounds, we guarantee that the generalization error of the SGD estimator is smaller than that of the GD estimator with high probability.

We also point out that our result might shed new light on deep learning (Belkin et al., 2018). By the state-of-the-art mathematical theory of Neural Networks (NN), kernel and/or nonparametric methods can approximate the functional space of neural networks, see for example the NTK theory (Jacot et al., 2018), and the Radon bounded variance space description for ReLU NN (Parhi & Nowak, 2021). Our technique might allow one to characterize the SGD solution path and show the generalization property in those problem settings.

**Paper organization**. The rest of the paper is as follows: In Section 2, we review some relevant literature; In Section 3, we give the formulation of the nonparametric regression and define the optimization problem. We also formalize the algorithm that is considered in this work and make assumptions to analyze the algorithms; In Section 4, we state our main theory on the directional bias of SGD/GD in nonparametric regression, where we include both the directional bias result and the implication of the directional bias for generalization. Experiments are conducted to support our theory; In Section 5, we discuss the finding in this paper, and propose some future research topic. All the proof, experiment details, and more experiments are deferred to the appendix due to page limits.

## 2 LITERATURE REVIEW

In this section, we review some relevant works. For better understanding, we split into two subsections: Subsection 2.1 reviews the background of RKHS; Subsection 2.2 presents the state-of-the-art technique for analyzing the directional bias of the (S)GD algorithm.

### 2.1 RKHS

Kernel methods are among the core algorithms in machine learning and statistics (Bartlett et al., 2021). As proposed by Wahba (1990), the kernel method and RKHS serve as a unified framework of a group of nonparametric models, which extends the spline method. Later, kernel models become an important component in nonparametric models. In machine learning, the kernel-based method is always referred to as the "kernel trick". By lifting the $x$ variable to a high dimensional space via the kernel method, we can explore possibly nonlinear relationships between variables. Moreover, to

play the kernel trick, one can directly calculate the kernel function using original features. This is computationally efficient since we avoid calculating high dimension or infinite dimension features. For applications of kernel method in machine learning, one can see kernel regression for image processing (Takeda et al., 2007), for text mining (Greene & Cunningham, 2006), and for tasks in bioinformatics (Saigo et al., 2004).

Regarding deep learning, the kernel method is also important because it has implications for deep learning models. On the one hand, the kernel methods have similar benign overfitting behavior to neural network due to the implicit regularization and/or implicit bias phenomenon that we will review in the next subsection (Belkin et al., 2018). On the other hand, the RKHS itself is closely related to Neural Networks via the Neural Tangent Kernel theory (Jacot et al., 2018). This all indicates that to understand deep learning, one should first study kernel methods.

## 2.2 DIRECTIONAL BIAS

This paper analyzes the directional bias of SGD for the nonparametric regression. Directional bias, also referred to as implicit bias, of an algorithm refers to that its solution path is biased towards a certain direction. It works as that the algorithm prefers some directions over the others even though they may have the same objective function value. Since the algorithm selects a direction by itself, instead of explicitly required by any constraint, people use the term "implicit". It is worth noting that the *implicit regularization* is related to implicit bias. The implicit regularization refers to that the converged point of an algorithm is like a regularized estimator, even if the objective function is not explicitly regularized. One can also interpret implicit regularization as the "final result" of implicit bias. In the recent work by Derezinski et al. (2020), implicit regularization is used to develop an exact bound for double descent in linear regression. In this way, implicit regularization/bias serves as a way to explain some deep learning phenomenons that could not be addressed by the classical empirical risk minimization (ERM) framework. Therefore, it is important to study directional bias.

State-of-the-art study on the directional bias of first-order algorithms can be divided into two categories by the technique they use:

The first category is the (stochastic) gradient flow method, by taking an infinitesimal step size in (S)GD, the parameter dynamic follows a (stochastic) differential equation. Studying the solution path and the stationary point of the underlying differential equation helps to reveal the property of the parameter estimation. We list some works that use the first method to show the directional bias result of the (stochastic) gradient descent. Liu et al. (2018) analyze the Momentum SGD (MSGD) with infinitesimal step size, and show that the solution path escapes from the saddle for a nonconvex objective function. It is worth noting that in their case, the associated stochastic differential equation defines a complicated stochastic process, thus they replace it with an appropriate diffusion process, and the analysis is done based on such diffusion approximation. If one analyzes a stochastic gradient flow and finds it intractable, one may consider using the technique of diffusion approximation. Ali et al. (2020) shows the stochastic gradient flow for the linear regression problem $\min_{\boldsymbol{w}} \|X\boldsymbol{w} - \mathbf{y}\|_2^2$ has a solution path close to the solution path of Ridge regression; Blanc et al. (2020) shows the stochastic gradient flow for a general loss function has a solution path close to the solution path of gradient flow on the objective function plus some extra penalty terms, and they explicitly identify the penalty terms; Smith et al. (2021) go one more step from the infinitesimal step size to small step size, and characterize the effect of small step size as an extra penalty term in the gradient flow.

Another category is analyzing the discrete (S)GD sequence. This technique in general just requires a moderate step size such that the algorithm converges (or nearly converges), thus it is more meaningful from a practical perspective. We also find some directional bias work that is based on this technique. Vaskevicius et al. (2019); Zhao et al. (2019); Fan et al. (2021) analyze Hadamard reparameterized GD in sparse regression. They divide the true parameter into strong, weak, and $0$ parts, and for each part, they carefully develop the stepwise error bound for each step of GD. They finally show that an early-stopped estimator along the solution path achieves the minimax optimal error rate for sparse regression, which indicates that the solution path is in the direction that biased towards a sparse solution. Recently, Wu et al. (2021) show that for overparameterized linear regression, SGD with moderate step size converges to the minimum norm interpolant in the direction that corresponds to the largest eigenvalue of the design matrix, while GD converges in the direction that corresponds to the smallest eigenvalue. For Neural Networks in the 'lazy training' regime, **?** shows

that GD also converges in the direction of the smallest eigenvalue of the Neural Tangent Kernel. Their result further reveal the mechanism of the directional bias as: GD fits the direction of a larger eigenvalue faster at the beginning of the training, left the smaller eigenvalue direction unfitted; later the direction of smaller eigenvalue is fitted, resulting in that the estimator goes in this direction.

## 3 PROBLEM FORMULATION

We give our problem formulation in this section. In Subsection 3.1 we define the kernel regression model and objective function; in Subsection 3.2, we give the SGD and GD algorithms; in Subsection 3.3, we state our assumption for later analysis. Due to the page limit, details of the nonparametric regression, RKHS and justifications for the assumption are deferred to Appendix A and B.

### 3.1 KERNEL REGRESSION

Suppose we are given $n$ data pairs $\{\boldsymbol{x}_i, y_i\}_{i=1}^n$ generated from an unknown model $y = f(\boldsymbol{x})$, where $\boldsymbol{x}_i \in \mathcal{X} \subset \mathcal{R}^p$ and $y_i \in \mathcal{R}$. The goal is to estimate the unknown model $f$ from the data. To achieve the goal, one way is to find an $f$ that minimizes the empirical risk function

$$\min_f \frac{1}{n} \sum_{i=1}^n \ell(y_i, f(\boldsymbol{x}_i)) \tag{1}$$

where $\ell$ is the loss function. For the regression task, we use the squared loss $\ell(y, \boldsymbol{x}) = \frac{1}{2}(y - f(\boldsymbol{x}))^2$.

One can see that problem (1) is not well-defined, as there are infinitely many solutions to $\forall i : f(\boldsymbol{x}_i) = y_i$, and some of them do not generalize for a new test data. One way to fix it is to restrict $f \in \mathcal{H}$ and add regularization term in $\|f\|_{\mathcal{H}}$ to problem (1) for smoothness, where $\mathcal{H}$ is a RKHS with reproducing kernel $K(\cdot, \cdot)$ and $\|\cdot\|_{\mathcal{H}}$ is the Hilbert norm. Adding these restrictions and applying Representer Theorem, problem (1) with the squared loss becomes

$$\min_{\boldsymbol{\alpha} \in \mathcal{R}^n} \frac{1}{2n} \sum_{i=1}^n (y_i - \boldsymbol{K}_i^T \boldsymbol{\alpha})^2 \tag{2}$$

$$= \frac{1}{2n} \|\boldsymbol{y} - K\boldsymbol{\alpha}\|_2^2$$

where $\boldsymbol{K}_i^T$ is the $i$th row of $K := K(X, X) = (K(\boldsymbol{x}_i, \boldsymbol{x}_j))_{i,j}$. For a parameter $\boldsymbol{\alpha}$, the corresponding estimator in $\mathcal{H}$ is $f(\cdot) = \sum_{i=1}^n \alpha_i K(\boldsymbol{x}_i, \cdot) := \boldsymbol{\alpha}^T K(\cdot, X)$.

Now when $K$ is invertible, it is trivial that any algorithm on objective function (2) (if it converges) converges at the unique minimal, that is, $\hat{\boldsymbol{\alpha}} = K(X, X)^{-1} \boldsymbol{y}$, result in the RKHS functional estimator

$$\hat{f}(\boldsymbol{x}) = K(\boldsymbol{x}, X)^T K(X, X)^{-1} \boldsymbol{y} \tag{3}$$

where $K(\boldsymbol{x}, X)^T = (K(\boldsymbol{x}, \boldsymbol{x}_1), \ldots, K(\boldsymbol{x}, \boldsymbol{x}_n))$. Estimator (3) is the minimum norm interpolant as given by following problem:

$$\arg \min_{f \in \mathcal{H}} \{\|f\|_{\mathcal{H}} : f(\boldsymbol{x}_i) = y_i, i = 1, \ldots, n\}$$

And its property has been studied in Liang & Rakhlin (2020).

In this paper, we compare the convergence direction of SGD and GD to $\hat{\alpha}$. Specifically, we consider a two-stage SGD with a phase transition from a larger step size to a decreased step size. Note that this matches the training scheme people always use in practice for SGD algorithms: decreasing the step size after training for a few epochs. For that purpose, in the following sections, we define the one-step SGD/GD update and state our assumptions and notations for analysis.

### 3.2 ONE STEP SGD/GD UPDATE

In this paper, we consider the SGD algorithm as follows. For objective function (2), denote the parameter estimation at $t$th step as $\alpha_t$, then SGD update $\boldsymbol{\alpha}_{t+1}$ as

$$\boldsymbol{\alpha}_{t+1} = \boldsymbol{\alpha}_t - \eta_t (\boldsymbol{K}_{i_t}^T \boldsymbol{\alpha}_t - y_{i_t}) \cdot \boldsymbol{K}_{i_t} \tag{4}$$

where $i_t$ is uniformly random sampled from $\{1, \ldots, n\}$.

The GD update $\boldsymbol{\alpha}_{t+1}$ as

$$\boldsymbol{\alpha}_{t+1} = \boldsymbol{\alpha}_t - \frac{\eta_t}{n} K^T (K\boldsymbol{\alpha}_t - \boldsymbol{y}) = \boldsymbol{\alpha}_t - \frac{\eta_t}{n} K (K\boldsymbol{\alpha}_t - \boldsymbol{y}) \qquad (5)$$

### 3.3 ASSUMPTIONS AND NOTATIONS

We state our assumption on the gram matrix in a unified format. Later we show in Appendix B that some popular kernel families satisfy our assumption.

**Assumption 1** (Diagonal Dominant gram matrix). *Denote by $K = K(X, X)$ the gram matrix, we assume that $K$ is diagonal dominant. Specifically, suppose w.l.o.g. that $K_{1,1} \geq K_{2,2} \geq \ldots \geq K_{n,n} > 0$, then we have for a small value $\tau$ that*

$$|K_{i,j}| \leq \tau \ll K_{n,n}, \forall i \neq j$$

**Remark 2.** *Diagonal dominant gram matrix is common in kernel learning. Mathematically, one can justify that a gram matrix is diagonal dominant by imposing proper assumptions on the kernel function $K(\cdot, \cdot)$ and the data distribution. The following proposition shows diagonal dominance for bilinear kernel. We can check for some other popular kernel to be diagonal dominant, which we defer to Appendix B due to the page limit.*

**Remark 3.** *Think of the kernel function as the inner product of high-dimensional features, the resulting gram matrix is diagonal dominant when the high-dimension features are sparse. This happens for a lot of practical problems (Schölkopf et al., 2002; Weston et al., 2003), for example, when linear or string kernels are applied to text data (Greene & Cunningham, 2006), when domain-specific kernels are applied to image retrieval (Tao et al., 2004) and bioinformatics (Saigo et al., 2004), and when the Global Alignment kernel is applied to most datasets (Cuturi et al., 2007; Cuturi, 2011).*

**Proposition 4** (Lemma 1 in Wu et al. (2021)). *Consider the bilinear kernel $K(\boldsymbol{x}, \boldsymbol{x}') := \langle \boldsymbol{x}, \boldsymbol{x}' \rangle$. Assume the data $\boldsymbol{x}_i, i = 1, \ldots, n$ are i.i.d. uniformly distributed on the unit sphere $S^{d-1}$, where $d \gg n$. When $d \geq 4 \log(2n^2/\delta)$ for some $\delta \in (0, 1)$. Then with probability at least $1 - \delta$, we have*

$$|K_{i,j}| = |\langle \boldsymbol{x}_i, \boldsymbol{x}_j \rangle| < \tilde{\tau} := \tilde{\mathcal{O}}(\frac{1}{\sqrt{d}}) \ll K_{n,n} = 1, \forall i \neq j.$$

It is meaningful to note that the diagonal dominance is undesired in classification and clustering tasks. It indicates that the data pieces are dissimilar to each other as measured by the kernel function, and thus generates very little information for classification/clustering. One may find a lot of works on solving the issue of diagonal dominance in these cases, for example, Greene & Cunningham (2006); Kandola et al. (2003). But for the regression task, the diagonal dominance, in other words, the dissimilarity of data points, may have benefits. One can find similar conditions such as Restricted Isometry Property and $s-$goodness that describes linearly dissimilar features in a huge regression literature as Candes & Tao (2007); Candès (2008); Chen & Donoho (1994). Such conditions are required for proving minimax optimality or exact recovery of a sparse signal in sparse settings. In our case, we adopt the dissimilarity concept and apply it to data points in high-dimensional nonlinear feature space. Later we will see that in the existence of diagonal dominance, the directional bias drives SGD to select a good solution that generalizes well among all solutions of the same level of empirical loss. In this way, our SGD estimator benefits from the diagonal dominance.

**Notations**. We use the following notations throughout the remaining of this work. For the gram matrix $K$, denote $K_{i,j}$ be the element at $i$th row $j$th column of $K$. Denote $\lambda_i = K_{i,i} = K(x_i, x_i)$, and assume w.l.o.g. that $\lambda_1 \geq \lambda_2 \geq \ldots \geq \lambda_n$. Denote the $i$th column of $K$ as $\boldsymbol{K}_i$, let $K_{-1} = [K_2, \ldots, K_n]$. Assume $K$ is full rank, denote $P_{-1}$ the projection onto column space of $K_{-1}$, and $P_1 = I - P_{-1}$. And denote $\gamma_1 \geq \ldots \geq \gamma_n > 0$ eigenvalues of $K$ in non-increasing order.

## 4 MAIN RESULT

The main results are presented in two subsections: Subsection 4.1 states the different directional bias result of SGD and GD estimators; Subsection 4.2 shows that directional bias towards a certain

directional leads to good generalization performance, and further applies this result to show that SGD generalizes better than GD.

## 4.1 DIRECTIONAL BIAS

Since we assumed that $K$ is full rank, then SGD and GD algorithm on objective function (2) converges to $\hat{\boldsymbol{\alpha}} = K^{-1}\boldsymbol{y}$ (when they converge). We are interested in the direction at which $\boldsymbol{\alpha}_t$ converges to $\hat{\boldsymbol{\alpha}}$, i.e. the quantity

$$\boldsymbol{b}_t := \boldsymbol{\alpha}_t - \hat{\boldsymbol{\alpha}}$$

With assumption 1 that the gram matrix is diagonal dominant, we prove that a two-stage SGD has $\boldsymbol{b}_t$ converge in the direction of the largest eigenvector of $K$.

**Theorem 5** (Direction bias of SGD estimator). *Assume Assumption 1 holds, run a two-stage SGD with a fixed step size for each stage: stage 1 with step size $\eta_1$ for steps $1, \ldots, k_1$, stage 2 with step size $\eta_2$ for steps $k_1 + 1, \ldots, k_2$, such that*

$$\frac{2}{\lambda_1^2 - C_1\sqrt{n}\tau} < \eta_1 < \frac{2}{\lambda_2^2 + C_2\sqrt{n}\tau}$$

$$\eta_2 < \frac{1}{\lambda_1^2 + C_3\sqrt{n}\tau}$$

*where $C_1, C_2, C_3$ are constants that are specified in the Appendix E. For a small $\epsilon > 0$ such that $n\tau < poly(\epsilon)$ there exists $k_1 = \mathcal{O}(\log\frac{1}{\epsilon})$ and $k_2$ such that*

$$(1 - 2\epsilon)\gamma_1 \leq \frac{E[\|K\boldsymbol{b}_{k_2}^{SGD}\|_2]}{E[\|\boldsymbol{b}_{k_2}^{SGD}\|_2]} \leq \gamma_1$$

*That is, $\boldsymbol{b}_{k_2}^{SGD}$ is close to the direction of the largest eigenvector of $K$.*

**Remark 6.** *One should assume $\tau$ in Assumption 1 to be small enough for $\epsilon$ to be very small if one would like the resulting estimator $\boldsymbol{b}_{k_2}^{SGD}$ to have the direction that corresponds to the largest eigenvalue of $K$. Later we will see that if one only wants different directional bias of SGD and GD estimators, a moderate $\epsilon$ is allowed and then the assumption on $\tau$ is not that strong.*

The proof of Theorem 5 is in Appendix E. Next, we see that GD has $\boldsymbol{b}_t$ converge in the direction of the smallest eigenvector of $K$, which contrasts with the directional bias of SGD.

**Theorem 7** (Direction bias of GD estimator). *Assume Assumption 1 hold, run GD with a fixed step size:*

$$\eta < \frac{n}{(\lambda_1 + n\tau)^2},$$

*for a small $\epsilon' > 0$, run $k = \mathcal{O}(\log\frac{1}{\epsilon'})$ steps of GD, we have*

$$\gamma_n \leq \frac{\|K\boldsymbol{b}_k^{GD}\|_2}{\|\boldsymbol{b}_k^{GD}\|_2} \leq \sqrt{1 + \epsilon'}\gamma_n$$

*That is, $b_t$ is close to the direction of the smallest eigenvector of $K$.*

**Remark 8.** *The assumption (on $\tau$) is mild for differentiating the directional bias of SGD and GD. Comparing Theorem 5 and 7, we see that as long as $\gamma_n < (1 - 2\epsilon)\gamma_1$, by taking $k$ large enough we always have*

$$\frac{\|K\boldsymbol{b}_k^{GD}\|_2}{\|\boldsymbol{b}_k^{GD}\|_2} < \frac{E\|K\boldsymbol{b}_{k_2}^{SGD}\|_2}{E\|\boldsymbol{b}_{k_2}^{SGD}\|_2}$$

*That is, one may expect $\boldsymbol{b}_{k_2}^{SGD}$ to be in the direction of larger eigenvalue compared with $\boldsymbol{b}_k^{GD}$. In the following subsection, we see that the directional bias towards a larger eigenvalue of the kernel is good for generalization, which leads to our title that directional bias helps SGD to generalize in kernel regression.*

The proof of Theorem 7 is in Appendix F. Although there is assumption 1 in Theorem 7, it is just used to bound the step size so that GD converges; the diagonal dominant structure of $K$ is not required for the directional bias for GD to hold. Moreover, the choice of $\epsilon'$ is independent of the assumption on $\tau$, then for an arbitrarily small $\epsilon' > 0$, we can always run GD long enough such that the theorem holds and the estimator $\boldsymbol{b}_k^{GD}$ is arbitrarily close to the smallest eigenvector.

### 4.2 EFFECT OF DIRECTIONAL BIAS

In this subsection, the estimator that has a directional bias towards the largest eigenvalue of the Hessian is shown to give the best parameter estimation error among all estimators that have the same squared in-sample loss, see Theorem 9. Later we define a realizable problem setting of kernel regression where the generalization error depends on a term similar to the parameter estimation error, and in this way, the directional bias helps SGD to generalize.

**Theorem 9.** *Consider approximately minimizing the quadratic loss*

$$L(\boldsymbol{w}) = \|A\boldsymbol{w} - \boldsymbol{y}\|_2^2$$

*Assume there is a ground truth $\boldsymbol{w}^*$ such that $\boldsymbol{y} = A\boldsymbol{w}^*$, for a fixed level of the quadratic loss, the parameter estimation error $\|\boldsymbol{w} - \boldsymbol{w}^*\|_2^2$ has a lower bound*

$$\forall \boldsymbol{w} \in \{\boldsymbol{w} : L(\boldsymbol{w}) = a\} : \|\boldsymbol{w} - \boldsymbol{w}^*\|_2^2 \geq \frac{a}{\|A^T A\|_2}$$

*Moreover, the equality is obtained when $\boldsymbol{w} - \boldsymbol{w}^*$ is in the direction of the eigenvector that corresponds to the largest eigenvalue of $A^T A$.*

**Remark 10.** *Theorem 9 implies that the directional bias towards the largest eigenvalue is good for parameter estimation. As discussed in Remark 8, the SGD estimator is biased towards a larger eigenvalue compared to the GD estimator, then by Theorem 9 the SGD estimator better estimates the true parameter and thus generalizes better. We formalize this statement in the following paragraphs.*

The proof of Theorem 9 is in Appendix G.1.

Suppose $\exists f^* \in \mathcal{H}$ such that $y = f^*(x)$. Consider the generalization error $L_D(f) := \|f - f^*\|_{\mathcal{H}}^2$, for an algorithm output $f^{alg}$, we decompose its generalization error as:

$$L_D(f^{alg}) - \inf_{f \in \mathcal{H}} L_D(f) = \underbrace{L_D(f^{alg}) - \inf_{f \in \mathcal{H}_s} L_D(f)}_{:=\Delta(f^{alg}), \text{ estimation error}} + \underbrace{\inf_{f \in \mathcal{H}_s} L_D(f) - \inf_{f \in \mathcal{H}} L_D(f)}_{\text{approximation error}}$$

where $\mathcal{H}_s$ is the hypothesis class that the output of the algorithm is restricted to. By formulation (2), we have our hypothesis class as

$$\mathcal{H}_s = \{f \in \mathcal{H} : f = \boldsymbol{\alpha}^T K(\cdot, X), \boldsymbol{\alpha} \in \mathcal{R}^n\}$$

We define the $a-$level set of training loss:

$$\nu_a = \{f \in \mathcal{H}_s : f = \boldsymbol{\alpha}^T K(\cdot, X), \frac{1}{2n}\|K\boldsymbol{\alpha} - \boldsymbol{y}\|_2^2 = a\},$$

denote $\Delta_a^* := \inf_{f \in \nu_a} \Delta(f)$.

Note that the approximation error can not be improved by choice of algorithm unless we change the hypothesis class, which is, changing the problem formulation in our case. So we just minimize the estimation error for estimators that are in the $a-$level set. As shown in Appendix G.2, one can check the estimation error is given by

$$f \in \mathcal{H}_s : \Delta(f) = \boldsymbol{b}^T K \boldsymbol{b}$$

where $\boldsymbol{b} = \boldsymbol{\alpha} - \hat{\boldsymbol{\alpha}}$. By similar reasoning as Theorem 9, the estimation error is minimized when $\boldsymbol{b}$ is in the direction of the largest eigenvalue of $K$, so the directional bias towards a larger eigenvalue helps to generalize in kernel regression. We formalize the statement for comparing the estimation error of SGD and GD in the following theorem.

**Theorem 11** (Generalization performance). *Follow Theorems 5 and 7, we have the following:*

- *The output of SGD has $E[\Delta^{1/2}(f^{SGD})] \leq (1 + 4\epsilon)(\Delta_a^*)^{1/2}$, where $a$ is such that $E[\|K\boldsymbol{\alpha}^{SGD} - y\|_2]^2 = 2na$ and $\epsilon$ could be any positive small constant;*

- *The output of GD has $\Delta(f^{GD}) \geq M\Delta_a^*$, where $a$ is the training loss of GD estimator, and $M = \frac{\gamma_1}{\gamma_n}(1 - \epsilon') > 1$ is a large constant.*

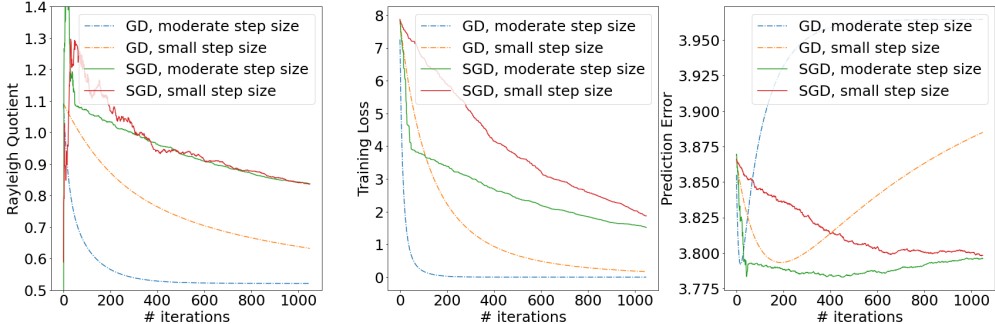

Figure 1: Kernel regression on synthetic data. We simulate data from a nonlinear regression model with Gaussian additive noise and fit kernel regression using a polynomial kernel. We run both SGD and GD for two step size schemes, see details in Appendix H.1. In the first plot, we show the directional bias by Rayleigh Quotient(RQ):= $\frac{\|K\boldsymbol{b}\|_2^2}{\|\boldsymbol{b}\|_2^2}$ (same as Theorem 5 and 7). The SGD indeed converges in the direction of a larger RQ, which matches our Theorems 5 and 7. In the third plot we show the prediction error of the solution paths, and the SGD does have lower prediction error than GD, even GD has smaller training loss than SGD. This supports Theorem 11.

**Remark 12.** *One read from the theorem that the SGD estimator is $(8\epsilon + 16\epsilon^2)-$optimum, while GD estimator is $(M-1)-$sub-optimum. Combine with Theorem 7 that $\epsilon' \overset{k\to\infty}{\longrightarrow} 0$, we can take $\epsilon$ in Theorem 5 such that $(1+4\epsilon)^2 < \gamma_1/\gamma_n$ to have $\Delta(f^{SGD}) < \Delta(f^{GD})$ with high probability. This finishes our claim that the SGD estimator generalize better than the GD estimator.*

The detailed proof of Theorem 11 is left to Appendix G.2.

**Numeric Study.** Figure 1 shows the simulation results of kernel regression, Figure 2 shows the results of a small ResNet-like Neural Network on FashionMNIST data (Xiao et al., 2017). Figure 1 and Figure 2a supports the directional bias results in Theorems 5 and 7, and Figure 2b validates Theorem 11. For details of the experiments and more experiments, see Appendix H.

**Remark 13.** *The purpose of experiment using a Neural Network (Figure 2) is as following: first, the Neural Network results support our finding on kernel regression, since Neural Network is related to kernel regression through NTK theory (Jacot et al., 2018); second, our experiment indicates that our result may be empirically true for the more general deep learning framework (Belkin et al., 2018), since this experiment uses a complicated network that may not be simply explained by the kernel method.*

## 5 DISCUSSION AND FURTHER WORK

Our work takes one more step towards understanding the directional bias of SGD in kernel learning. Here we discuss some implications of our results to deep learning.

**Implication for SGD scheme**: Our result shows the directional bias applies to SGD with annealing step size. Specifically, the first stage of SGD with moderate step size should run long enough, then in the second stage by decreasing step size we have the directional bias towards the largest eigenvalue of the Hessian, which helps in benign overfitting. This explains a technique for tuning the learning rate that people use in practice: start with a large step size, run long enough until the error plateaus, then decrease the step size (He et al., 2016). Although this technique is always used to speed convergence, we show that it also helps in benign overfitting, which becomes even better.

**Implication for deep learning**: Our assumption for analysis implies certain structures for deep learning models. Per our examples in Appendix B and our discussion in Remark 3, our assumption holds when the feature space is high dimensional and/or when features are possibly sparse. This matches the deep learning scenario where we have a highly overparameterized model and when the trained parameter estimator becomes sparse. Besides, considering that some deep learning tasks can

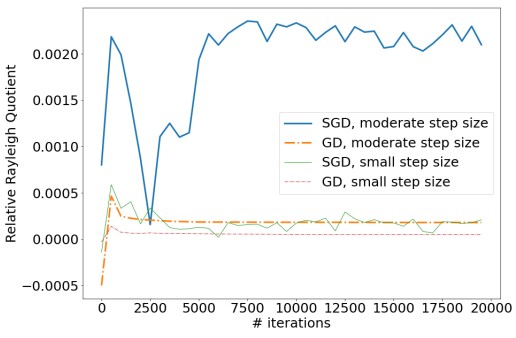
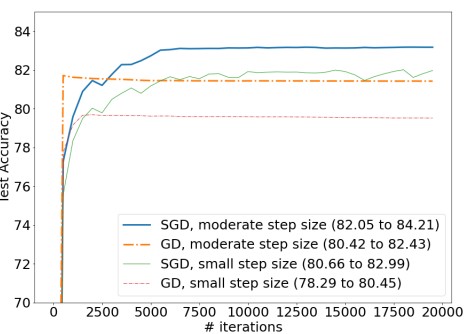

(a) Relative Rayleigh Quotient.     (b) Test accuracy on FashionMNIST

Figure 2: The experiment of a small ResNet-like Neural Network on FashionMNIST. In (a), we follow Wu et al. (2021) to use the Relative Rayleigh Quotient(RRQ) as the measurement of the convergence direction, where higher RRQ means that the convergence direction is closer to the larger eigenvalue direction of the Hessian. The SGD with moderate step size has higher RRQ than the GD with either moderate step size or small step size, which supports the theory in Theorems 5 and 7. It is also interesting to observe SGD with a small step size also has a different directional bias compared with SGD with a moderate step size. In (b), we plot the testing accuracy from 20 repetitions of experiments, the test accuracy (inside bracket) of SGD with moderate step size is higher than the other cases, and we have Wilcoxon signed-rank test to confirm that the difference is significant at 0.01 level. The test accuracy validates Theorem 11. For more details of the experiments, the rank test, and more experiments, see Appendix H.2.

be approximated by kernel learning (Jacot et al., 2018), our results help in explaining why the SGD estimator can benign overfitting in an overparameterized deep learning.

Just as stated in Belkin et al. (2018), to understand deep learning one needs to understand kernel learning. This work takes a step in understanding kernel learning, and we expect more steps that go beyond this work towards understanding deep learning, possibly for some complicated structure that could not be approximated by kernel learning.

**Reproducibility Statement:** For all theoretical results presented in this paper, we carefully state and justify the assumption, we also include the proof in Appendix. For all experiments, we state the implementation details in Appendix, and we include the necessary code and data for reproducing our result in the supplementary material.

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

# A  BACKGROUND ON RKHS

This section details background on RKHS in two subsections. The first subsection includes notations, theorems, and an example of RKHS, the second section reduces the kernel regression in RKHS from infinite dimension to finite dimension, which gives our objective function (2) .

## A.1  NONPARAMETRIC MODEL IN RKHS

In this subsection, we give the definition and notations for our model in RKHS, and its associated norms, basis, etc. The definitions are similar to those in Raskutti et al. (2012).

Given $n$ data pairs $\{\boldsymbol{x}_i, y_i\}_{i=1}^n$, where $\boldsymbol{x}_i \in \mathcal{X} \subset \mathcal{R}^p$ and $y_i \in \mathcal{R}$, assume that $y_i$s are associated with $\boldsymbol{x}_i$s through $f(\boldsymbol{x}_i)$, where $f(\cdot)$ is some unknown function in the reproducing kernel Hilbert space (RKHS) of functions $\mathcal{X} \to \mathcal{R}$, our goal is to estimate the function $f(\cdot)$ from the data.

Denote the RKHS where $f$ lives as $\mathcal{H}$, with reproducing kernel $K : \mathcal{X} \times \mathcal{X} \to \mathcal{R}_+$ (which is known to us). And we associate the functions in $\mathcal{H}$ with probability measure $\mathbb{Q}$, assume w.l.o.g. that $\int_{\mathcal{R}^p} f(\boldsymbol{a})d\mathbb{Q}(\boldsymbol{a}) = 0$. By Mercer's theorem, $K$ has eigen-expansion:

$$K(\boldsymbol{a}, \boldsymbol{b}) = \sum_{j=1}^\infty \gamma_j \phi_j(\boldsymbol{a})\phi_j(\boldsymbol{b})$$

Where $\{\phi_j\}_{j=1}^\infty$ are orthonormal basis in $\mathbb{L}^2(\mathbb{Q})$, w.r.t. the usual inner product in $\mathbb{L}^2(\mathbb{Q})$ as

$$\langle g(\cdot), h(\cdot)\rangle_{\mathbb{L}^2(\mathbb{Q})} = \int_{\mathcal{X}} g(\boldsymbol{a})h(\boldsymbol{a})d\mathbb{Q}(\boldsymbol{a})$$

Now for any $f \in \mathcal{H}$, we can expand $f(\cdot) = \sum_{j=1}^\infty c_j\phi_j(\cdot)$, where $c_j = \langle f(\cdot), \phi_j(\cdot)\rangle_{\mathbb{L}^2(\mathbb{Q})}$. And for $f(\cdot) = \sum_{j=1}^\infty c_j\phi_j(\cdot), g(\cdot) = \sum_{j=1}^\infty c_j'\phi_j(\cdot)$, by Parseval's theorem

$$\langle f(\cdot), g(\cdot)\rangle_{\mathbb{L}^2(\mathbb{Q})} = \sum_{j=1}^\infty c_j c_j'$$

And we have another inner product that is defined for RKHS $\mathcal{H}$ as

$$\langle f(\cdot), g(\cdot)\rangle_{\mathcal{H}} = \sum_{j=1}^\infty \frac{c_j c_j'}{\gamma_j}$$

The reproducing property of RKHS says that $\forall f \in \mathcal{H}$, we have

$$\langle f(\cdot), K(\cdot, \boldsymbol{x})\rangle_{\mathcal{H}} = f(\boldsymbol{x})$$

**Cubic Splines Formulate a RKHS**. We go over an example of RKHS for better understanding. Consider the cubic spline of one dimension, we can show that the space of cubic splines is a RKHS. One can also find the cubic spline example in Hazimeh et al. (2021). For more details on the relationship between polynomial smoothing splines and RKHS, one can check Section 1.2 of Wahba (1990) .

Assume w.l.o.g. that $x_i \in \mathcal{X} = [0, 1] \subset \mathcal{R}$. The cubic spline $f$ on $\mathcal{X}$ is continuous, has a continuous first-order derivative and square integrable second order derivative. By Taylor's theorem with remainder, we have

$$f(t) = f(0) + tf'(0) + \int_0^t (t-u)f''(u)du$$

$$= f(0) + tf'(0) + \int_0^1 (t-u)_+ f''(u)du$$

where $(t-u)_+ = \max\{0, t-u\}$. Let $\mathcal{B}$ be the set of cubic splines $f$ on $[0, 1]$ that satisfies the boundary condition $f(0) = f'(0) = 0$, then for $f \in \mathcal{B}$

$$f(t) = \int_0^1 (t-u)_+ f''(u)du$$

Let $G(t, u) = (t - u)_+$, then we claim that $\mathcal{B}$ is RKHS with reproducing kernel

$$K(s, t) = \int_0^1 G(s, u)G(t, u)du$$

and inner product

$$\langle f, g \rangle_{\mathcal{B}} = \int_0^1 f''(u)g''(u)du$$

as one can check the reproducing property

$$\langle f(\cdot), K(\cdot, t) \rangle_{\mathcal{B}} = \int_0^1 \frac{\partial^2 K(u, t)}{\partial u^2} f''(u)du$$

$$= \int_0^1 (t - u)_+ f''(u)du = f(t)$$

## A.2 Optimization problem considered

This subsection gives problem formulation of kernel regression. Given data pairs $\{x_i, y_i\}_{i=1}^n$ and RKHS $\mathcal{H}$, consider a loss function $\ell$ which is selected according to how $y$ is connected with $f(x)$, we may estimate the model by

$$\min_{f \in \mathcal{H}} \frac{1}{n} \sum_{i=1}^n \ell(y_i, f(x_i)) \tag{6}$$

$$= \min_{c_j} \frac{1}{n} \sum_{i=1}^n \ell(y_i, \sum_{j=1}^\infty c_j \phi_j(x_i))$$

Example for $\ell$ includes

- Squared error loss $\ell(y, f) = (y - f)^2$, which is usually used in regression;
- 0-1 loss $\ell(y, f) = \mathbf{1}(y * f > 0)$, for binary classification;
- Logistic loss $\ell(y, f) = \log(1 + \exp(-y * f))$, also a loss function for classification, can be considered as a surrogate function of 0-1 loss, and is the same as negative log likelihood function in logistic regression.

Let us come back to the nonparametric model part, to control the model smoothness, the usual practice is to add a penalty to objective (6), result in

$$\min_{f \in \mathcal{H}} \sum_{i=1}^n \ell(y_i, f(x_i)) + \lambda \text{pen}(f)$$

A popular choice of $\text{pen}(f)$ is $\|f\|_{\mathcal{H}}^2$, or any strictly increasing function of $\|f\|_{\mathcal{H}}^2$. Such method is explicitly controlling the model smoothness, and by Representer Theorem, it has solution of the form

$$f(\cdot) = \sum_{i=1}^n \alpha_i K(\cdot, x_i) \tag{7}$$

Plug equation (7) into objective (6), we have the problem becomes

$$\min_{\alpha_{i'}} \frac{1}{n} \sum_{i=1}^n \ell(y_i, \sum_{i'=1}^n \alpha_{i'} K(x_i, x_{i'}))$$

Which gives the formulation (2) under loss function $\ell(y, f(x)) = (y - f(x))^2/2$.

## B    DIAGONAL DOMINANCE OF SOME POPULAR KERNELS

In this section, we justify Assumption 1 by figuring out a problem setting where some popular kernels give a diagonal dominant gram matrix. For simplicity, we assume the following data distribution throughout this section:

$$\mathbf{x}_i \in R^d, i = 1, \ldots, n, \text{ are normalized such that } \|\boldsymbol{x}_i\|_2^2 = 1; \tag{A.1}$$

$$\text{The direction of } \boldsymbol{x}_i\text{s are i.i.d. uniformly distributed on the unit sphere } S^{d-1}; \tag{A.2}$$

$$d \gg n \text{ (overparameterized setting).} \tag{A.3}$$

Given assumption set (A), we can bound the inner product of data $\langle \boldsymbol{x}_i, \boldsymbol{x}_j \rangle$ with high probability as follows:

**Lemma 14** (Lemma 1 in Wu et al. (2021)). *Under assumption set* (A), *let* $d \geq 4\log(2n^2/\delta)$ *for some* $\delta \in (0, 1)$. *Then with probability at least* $1 - \delta$, *we have*

$$|\langle \boldsymbol{x}_i, \boldsymbol{x}_j \rangle| < \tilde{\tau} := \tilde{\mathcal{O}}(\frac{1}{\sqrt{d}}), \forall i \neq j$$

*Proof.* See proof of Lemma 1 in Wu et al. (2021). □

The bound on the inner product $\langle \boldsymbol{x}_i, \boldsymbol{x}_j \rangle$ induces bound on $K(\boldsymbol{x}_i, \boldsymbol{x}_j)$ for some popular kernels. We show the diagonal dominance for two groups of kernels in the following propositions, and list some examples for kernels in each group.

**Proposition 15** (Inner product kernel). *The inner product kernel is defined as a smooth transformation of inner product. We can write it as:*

$$K(\boldsymbol{x}_i, \boldsymbol{x}_j) = g(\langle \boldsymbol{x}_i, \boldsymbol{x}_j \rangle)$$

*Assume assumptions* (A) *hold, and assume the function* $g : [-1, 1] \rightarrow R$ *satisfies:*

$$g \text{ is convex}; \tag{B.1}$$

$$g \text{ is } L-\text{smooth, that is, } \nabla g \text{ is } L-\text{Lipschitz continuous}; \tag{B.2}$$

$$|g(0)| \leq c\tilde{\tau} \text{ for a constant } c, g'(0) \geq 0. \tag{B.3}$$

*we will have with probability* $1 - \delta$

$$|K_{i,j}| \leq (c + g'(0))\tilde{\tau} + \frac{L}{2}\tilde{\tau}^2 \text{ for } i \neq j \tag{10}$$

*where* $\delta$ *and* $\tilde{\tau}$ *the same as in Lemma 14. When* $g'(0)\tilde{\tau} + \frac{L}{2}\tilde{\tau}^2 \ll g(1)$ *for a small enough* $\tilde{\tau}$, *the gram matrix is diagonal dominant.*

*Proof.* We have the following with probability at least $1 - \delta$ by Lemma 14. For any off-diagonal elements of $K$:

$$\begin{aligned} K_{i,j} &= g(\langle \boldsymbol{x}_i, \boldsymbol{x}_j \rangle) \\ &\geq g(0) + g'(0)\langle \boldsymbol{x}_i, \boldsymbol{x}_j \rangle \\ &\geq -(g'(0) + c)\tilde{\tau} \end{aligned}$$

and

$$\begin{aligned} K_{i,j} &= g(\langle \boldsymbol{x}_i, \boldsymbol{x}_j \rangle) \\ &\leq g(0) + g'(0)\langle \boldsymbol{x}_i, \boldsymbol{x}_j \rangle + \frac{L}{2}\langle \boldsymbol{x}_i, \boldsymbol{x}_j \rangle^2 \\ &\leq (c + g'(0))\tilde{\tau} + \frac{L}{2}\tilde{\tau}^2 \end{aligned}$$

Thus $|K_{i,j}| \leq (c + g'(0))\tilde{\tau} + \frac{L}{2}\tilde{\tau}^2$. □

**Remark 16.** *We list some examples of inner product kernels that give diagonal dominant kernel matrices:*

- **Bilinear Kernel:** $K(\boldsymbol{x}, \boldsymbol{x}') = \langle \boldsymbol{x}, \boldsymbol{x}' \rangle$, then

$$|K(\boldsymbol{x}_i, \boldsymbol{x}_j)| \leq \tilde{\tau} \ll K(\boldsymbol{x}_n, \boldsymbol{x}_n) = 1.$$

- **Polynomial Kernel:** $K(\boldsymbol{x}, \boldsymbol{x}') = (\langle \boldsymbol{x}, \boldsymbol{x}' \rangle + c)^m$ for $m \in \mathbb{N}$ and $c \sim \mathcal{O}(\tilde{\tau})$, then by Proposition 15

$$|K(\boldsymbol{x}_i, \boldsymbol{x}_j)| \leq (1 + m)\tilde{\tau} + \frac{m * \exp((m-1)\tilde{\tau})}{2}\tilde{\tau}^2$$

when $(1+m)\tilde{\tau} + \frac{m*\exp((m-1)\tilde{\tau})}{2}\tilde{\tau}^2 \ll (1+c)^m$, we have diagonal dominant gram matrix.

- **Hyperbolic Tangent Kernel (Sigmoid Kernel):** $K(\boldsymbol{x}, \boldsymbol{x}') = \tanh(\alpha\langle \boldsymbol{x}, \boldsymbol{x}' \rangle + c)$, where $\alpha > 0, c \geq 0$. Note that Hyperbolic Tangent Kernel does not satisfy all the assumptions in Proposition 15, one can still calculate

$$|K(\boldsymbol{x}_i, \boldsymbol{x}_j)| \leq \tanh(\alpha\tilde{\tau} + c)$$

and

$$K(\boldsymbol{x}_k, \boldsymbol{x}_k) = \tanh(\alpha + c)$$

When $\tanh(\alpha\tilde{\tau} + c) \ll \tanh(\alpha + c)$ (which is the case when $\alpha$ is large, and $c, \tilde{\tau}$ are small enough), we have $|K(\boldsymbol{x}_i, \boldsymbol{x}_j)| \ll K(\boldsymbol{x}_n, \boldsymbol{x}_n)$ and the gram matrix is diagonal dominant.

**Proposition 17** (Radial Basis Function (RBF) kernel). *Radial Basis Function kernel depends on two data points through their distance, which is of following form*

$$K(\boldsymbol{x}_i, \boldsymbol{x}_j) = \exp(-\gamma\|\boldsymbol{x}_i - \boldsymbol{x}_j\|_2^2), \gamma > 0$$

*Assume assumptions (A) hold, when $\gamma = -c_0 \log(\tilde{\tau})$ for a constant $c_0$, we have with probability $1 - \delta$*

$$|K_{i,j}| \leq \tilde{\tau}^{2c_0(1-\tilde{\tau})} \ll K_{n,n} = 1, \text{ for } i \neq j$$

*where $\delta$ and $\tilde{\tau}$ the same as in Lemma 14. That is, the Gram matrix is diagonal dominant.*

*Proof.* Bound off-diagonal terms of $K$ by Lemma 14:

$$\begin{aligned} K_{i,j} &= \exp(-\gamma\|\boldsymbol{x}_i - \boldsymbol{x}_j\|_2^2) \\ &\leq \exp(2\gamma\tilde{\tau} - 2\gamma) \\ &= \tilde{\tau}^{2c_0(1-\tilde{\tau})} \end{aligned}$$

$\square$

**Remark 18.** *We note some popular kernels that are related to Radial Basis Function kernel, and show that they lead to diagonal dominance:*

- **Gaussian Kernel:** $K(\boldsymbol{x}, \boldsymbol{x}') = \exp(-\frac{\|\boldsymbol{x}-\boldsymbol{x}'\|_2^2}{2\sigma^2})$. *One can see that Gaussian Kernel is reparameterizing RBF Kernel by $\gamma = 1/(2\sigma^2)$. Thus Gaussian gram matrix is diagonal dominant when $\sigma^2 \sim \mathcal{O}(-\frac{1}{\log(\tilde{\tau})})$.*

- **Laplace Kernel:** $K(\boldsymbol{x}, \boldsymbol{x}') = \exp(-\frac{\|\boldsymbol{x}-\boldsymbol{x}'\|_2}{\sigma})$ *for $\sigma > 0$. The Laplace Kernel is very similar to Gaussian Kernel, and one can check by similar steps that when $\sigma \sim \mathcal{O}(-\frac{1}{\log(\tilde{\tau})})$, Laplace gram matrix is diagonal dominant.*

## C LEMMAS

This section includes two useful lemmas for characterizing the eigenvalues of a symmetric matrix.

**Lemma 19** (Gershgorin circle theorem, restated for symmetric matrix). *Let $A \in \mathcal{R}^{n \times n}$ be a symmetric matrix. Let $A_{ij}$ be the entry in the $i-$th row and the $j-$th column. Let*

$$R_i(A) := \sum_{j \neq i} |A_{ij}|, i = 1, \ldots, n$$

*Consider n Gershgorin discs*

$$D_i(A) := \{z \in \mathcal{R}, |z - A_{ii}| \leq R_i(A)\}, i = 1, \ldots, n$$

*The eigenvalues of A are in the union of Gershgorin discs*

$$G(A) := \cup_{i=1}^n D_i(A)$$

*Furthermore, if the union of k of the n discs that comprise $G(A)$ forms a set $G_k(A)$ that is disjoint from the remaining $n - k$ discs, then $G_k(A)$ contains exactly k eigenvalues of A, counted according to their algebraic multiplicities.*

*Proof.* See Horn & Johnson (2012), Chap 6.1, Theorem 6.1.1. □

**Lemma 20** (Cauchy interlacing theorem, restated for symmetric matrix). *Let $B \in \mathcal{R}^{m \times m}$ be a symmetric matrix, let $\boldsymbol{y} \in \mathcal{R}^n$ and $a \in \mathcal{R}$, and let $A = \begin{bmatrix} B & \boldsymbol{y} \\ \boldsymbol{y}^T & a \end{bmatrix}$. Then*

$$\lambda_1(A) \geq \lambda_1(B) \geq \lambda_2(A) \geq \ldots \geq \lambda_m(A) \geq \lambda_m(B) \geq \lambda_{m+1}(A).$$

*Proof.* See Horn & Johnson (2012), Chap 4.3, Theorem 4.3.17. □

## D  SPECTRUM OF GRAM MATRIX

This section analyzes the eigen structure of the gram matrix.

**Lemma 21** (Characterizing $K^2$). *Under Assumption 1, we have*

$$\langle \boldsymbol{K}_i, \boldsymbol{K}_i \rangle \in [\lambda_i^2, \lambda_i^2 + (n-1)\tau^2] \tag{11}$$
$$|\langle \boldsymbol{K}_i, \boldsymbol{K}_j \rangle| \leq [2\lambda_1 + (n-2)\tau]\tau, i \neq j \tag{12}$$

*Proof.* For $\langle \boldsymbol{K}_i, \boldsymbol{K}_i \rangle$

$$\langle \boldsymbol{K}_i, \boldsymbol{K}_i \rangle = K_{i,i}^2 + \sum_{l \neq i} K_{l,i}^2$$
$$\in [\lambda_i^2, \lambda_i^2 + (n-1)\tau^2]$$

And for $\langle \boldsymbol{K}_i, \boldsymbol{K}_j \rangle, i \neq j$

$$|\langle \boldsymbol{K}_i, \boldsymbol{K}_j \rangle| = |\sum_{l=1}^n K_{l,i} K_{l,j}|$$
$$= |K_{i,i} K_{i,j} + K_{j,i} K_{j,j} + \sum_{l \neq i,j} K_{l,i} K_{l,j}|$$
$$\leq |K_{i,i} K_{i,j}| + |K_{j,i} K_{j,j}| + \sum_{l \neq i,j} |K_{l,i} K_{l,j}|$$
$$\leq [\lambda_i + \lambda_j]\tau + (n-2)\tau^2$$
$$\leq [2\lambda_1 + (n-2)\tau]\tau$$

□

**Lemma 22** (Eigenvalue of $K$). *Under Assumption 1, we have*

$$\gamma_1 \leq \lambda_1 + n\tau \tag{13}$$
$$\gamma_n \geq \lambda_n - n\tau \tag{14}$$

*If we further assume $\lambda_j + n\tau < \lambda_{j-1} - n\tau$, we will have*

$$\gamma_{j-1} \geq \lambda_{j-1} - n\tau > \lambda_j + n\tau \geq \gamma_j.$$

*Proof.* Use Gershgorin circle theorem, calculate

$$R_i(K) = \sum_{j \neq i} |K_{i,j}| \leq n\tau$$

then

$$D_i(K) \subset [\lambda_i - n\tau, \lambda_i + n\tau].$$

By Gershgorin circle theorem, the lemma claim holds. $\qquad\square$

**Lemma 23** (Characterize $P_1 K$ and $P_{-1} K$)**.** *Recall our definition: $P_{-1}$ is the projection on column space of $K_{-1} = [\boldsymbol{K}_2, \boldsymbol{K}_3, \ldots, \boldsymbol{K}_n]$, and $P_1 = I - P_{-1}$. We claim the following hold*

$$P_1 K = [P_1 \boldsymbol{K}_1, \boldsymbol{0}, \ldots, \boldsymbol{0}] \tag{15}$$
$$P_{-1} K = [P_{-1} \boldsymbol{K}_1, \boldsymbol{K}_2, \ldots, \boldsymbol{K}_n] \tag{16}$$

*Assume $\tau$ is small enough that $n\tau \leq \mathcal{O}(1)$, $\lambda_n - n\tau \geq c_1 > 0$ and $\lambda_1 + n\tau \leq c_2$, let $c_3 := \frac{c_2}{c_1^2}$, then we have the following:*

$$\|P_{-1}\boldsymbol{K}_1\|_2 \in \left[0, c_3(2\lambda_1 + (n-2)\tau)\sqrt{n}\tau\right] \tag{17}$$
$$\|P_1\boldsymbol{K}_1\|_2 \in \left[\sqrt{\lambda_1^2 - c_3^2[2\lambda_1 + (n-2)\tau]^2 n\tau^2}, \lambda_1 + \sqrt{n}\tau\right]. \tag{18}$$

*Proof.* For $i \neq 1$, we calculate

$$P_{-1}\boldsymbol{K}_i = \boldsymbol{K}_i$$
$$P_1 \boldsymbol{K}_i = \boldsymbol{K}_i - P_{-1}\boldsymbol{K}_i = \boldsymbol{0}$$

thus we have equations (15) and (16).

For $\|P_{-1}\boldsymbol{K}_1\|_2$:

$$\begin{aligned}
&\|P_{-1}\boldsymbol{K}_1\|_2 \\
=&\|K_{-1}(K_{-1}^T K_{-1})^{-1}K_{-1}^T \boldsymbol{K}_1\|_2 \\
\leq&\|K_{-1}(K_{-1}^T K_{-1})^{-1}\|_2\|K_{-1}^T \boldsymbol{K}_1\|_2
\end{aligned}$$

where

$$\begin{aligned}
&\|K_{-1}^T \boldsymbol{K}_1\|_2^2 \\
=&\sum_{i=2}^n (\boldsymbol{K}_i^T \boldsymbol{K}_1)^2 \\
\overset{(12)}{\leq}&(n-1)[2\lambda_1 + (n-2)\tau]^2\tau^2 \\
\leq&[2\lambda_1 + (n-2)\tau]^2 n\tau^2
\end{aligned}$$

and $K_{-1}^T K_{-1}$ has all eigenvalues in $[\gamma_n^2, \gamma_1^2]$ by Cauchy interlacing theorem (Lemma 20), that is, all singular values of $K_{-1}$ are in $[c_1, c_2]$ by our assumption. Then

$$\|K_{-1}(K_{-1}^T K_{-1})^{-1}\|_2 \leq \frac{c_2}{c_1^2} := c_3$$

So we have

$$\begin{aligned}
\|P_{-1}\boldsymbol{K}_1\|_2 &\leq \|K_{-1}(K_{-1}^T K_{-1})^{-1}\|_2\|K_{-1}^T \boldsymbol{K}_1\|_2 \\
&\leq c_3[2\lambda_1 + (n-2)\tau]\sqrt{n}\tau.
\end{aligned}$$

For $\|P_1\boldsymbol{K}_1\|_2$:

$$\begin{aligned}
&\|P_1\boldsymbol{K}_1\|_2 \leq \|\boldsymbol{K}_1\|_2 \\
\leq&(\lambda_1^2 + (n-1)\tau^2)^{.5} \\
\leq&\lambda_1 + \sqrt{n}\tau
\end{aligned}$$

and

$$\|P_1 \boldsymbol{K}_1\|_2^2$$
$$= \|\boldsymbol{K}_1\|_2^2 - \|P_{-1}\boldsymbol{K}_1\|_2^2$$
$$\geq \lambda_1^2 - c_3^2[2\lambda_1 + (n-2)\tau]^2 n\tau^2.$$

$\square$

**Lemma 24** (Spectrum of $H_{-1} := P_{-1}KK^T P_{-1}$). *Assume*

$$c_3^2[2\lambda_1 + (n-2)\tau]^2 n\tau^2 + 2[2\lambda_1 + (n-2)\tau]n\tau \leq \lambda_n^2$$

*We have the following:*

- *$0$ is an eigenvalue of $H_{-1}$, corresponding eigenspace is the column space of $P_1$;*

- *Restricted to the column space of $P_{-1}$, the eigenvalues of $H_{-1}$ are all in the interval:*

$$\left(\lambda_n^2 - [2\lambda_1 + (n-2)\tau]n\tau, \lambda_2^2 + [2\lambda_1 + (n-1)\tau]n\tau\right).$$

*Proof.* The first claim is by construction of $P_1$ and $P_{-1}$.

For the second claim, note that $H_{-1}$ has the same eigenvalues as

$$H'_{-1} = (P_{-1}K)^T P_{-1}K$$

Now the diagonal entries of $H'_{-1}$ are:

$$(H'_{-1})_{ii} = \|P_{-1}\boldsymbol{K}_i\|_2^2 = \begin{cases} \|P_{-1}\boldsymbol{K}_1\|_2^2 \leq c_3^2[2\lambda_1 + (n-2)\tau]^2 n\tau^2 & , i = 1 \\ \|\boldsymbol{K}_i\|_2^2 \in [\lambda_i^2, \lambda_i^2 + (n-1)\tau^2] & , i \neq 1 \end{cases}$$

And the off-diagonal entries of $H'_{-1}$ are:

$$|(H'_{-1})_{ij}| = |\langle P_{-1}\boldsymbol{K}_i, P_{-1}\boldsymbol{K}_j \rangle| = |\langle \boldsymbol{K}_i, \boldsymbol{K}_j \rangle| \leq [2\lambda_1 + (n-2)\tau]\tau$$

To use Gershgorin circle theorem, calculate

$$R_i(H'_{-1}) = \sum_{j \neq i} |(H'_{-1})_{ij}| < [2\lambda_1 + (n-2)\tau]n\tau$$

Thus the Gershgorin discs:

$$D_1(H'_{-1}) \in (\|P_{-1}\boldsymbol{K}_1\|_2^2 - [2\lambda_1 + (n-2)\tau]n\tau, \|P_{-1}\boldsymbol{K}_1\|_2^2 + [2\lambda_1 + (n-2)\tau]n\tau)$$
$$D_i(H'_{-1}) \in (\|\boldsymbol{K}_i\|_2^2 - [2\lambda_1 + (n-2)\tau]n\tau, \|\boldsymbol{K}_i\|_2^2 + [2\lambda_1 + (n-2)\tau]n\tau)$$

when $c_3^2[2\lambda_1 + (n-2)\tau]^2 n\tau^2 + [2\lambda_1 + (n-2)\tau]n\tau \leq \lambda_n^2 - [2\lambda_1 + (n-2)\tau]n\tau$, the first Gershgorin discs does not intersect with the others, so we have $n-1$ nonzero eigenvalues in

$$\cup_{i=2}^n D_i(H'_{-1}) \subset \left(\lambda_n^2 - [2\lambda_1 + (n-2)\tau]n\tau, \lambda_2^2 + [2\lambda_1 + (n-1)\tau]n\tau\right).$$

$\square$

# E  DIRECTIONAL BIAS OF SGD WITH MODERATE STEP SIZE

This section gives formal proof of Theorem 5 and specifies the constants. The proof is done in four steps: Lemma 25 analyzes one update of SGD; Lemma 26 uses Lemma 25 to bound the first stage updates of SGD with moderate step size; Lemma 27 again uses Lemma 25, and bounds the second stage updates of SGD with small step size; finally, Theorem 28 combines Lemma 26 and Lemma 27 to formalize the directional bias of SGD, it is the same as Theorem 5, but restated using the constants defined therein.

**Lemma 25** (One step update of SGD). *Under Assumption 1, denote $A_t := E[\|P_1 \boldsymbol{b}_t\|_2]$, $B_t := E[\|P_{-1} \boldsymbol{b}_t\|_2]$, fix a constant $c_4 \geq (\lambda_1 + \sqrt{n}\tau)(2\lambda_1 + (n-2)\tau)c_3$, then we have:*

$$A_{t+1} \leq q_1(\eta) A_t + \xi(\eta) B_t \tag{19}$$
$$A_{t+1} \geq q_1(\eta) A_t - \xi(\eta) B_t \tag{20}$$
$$B_{t+1} \leq q_{-1}(\eta) B_t + \xi(\eta) A_t \tag{21}$$

*where*

$$q_1(\eta) = \frac{n-1}{n} + \frac{1}{n}|1 - \eta\|P_1 \boldsymbol{K}_1\|_2^2|$$

$$q_{-1}(\eta) = \sqrt{1 + \frac{\|KP_{-1}\boldsymbol{b}_t\|_2^2}{n\|P_{-1}\boldsymbol{b}_t\|_2^2}[\eta^2(\lambda_2^2 + (n-1)\tau^2) - 2\eta]}$$

$$\xi(\eta) = c_4 \eta n^{-1/2} \tau.$$

*Proof.* One step of SGD update is:
$$\boldsymbol{b}_{t+1} = \boldsymbol{b}_t - \eta \boldsymbol{K}_i \boldsymbol{K}_i^T \boldsymbol{b}_t = [I - \eta \boldsymbol{K}_i \boldsymbol{K}_i^T]\boldsymbol{b}_t$$
where $i$ is uniformly random sample from $[1, \ldots, n]$.

For inequalities (19) and (20), check
$$\begin{aligned}
P_1 \boldsymbol{b}_{t+1} &= P_1[I - \eta \boldsymbol{K}_i \boldsymbol{K}_i^T]\boldsymbol{b}_t \\
&= P_1 \boldsymbol{b}_t - \eta P_1 \boldsymbol{K}_i \boldsymbol{K}_i^T(P_1 + P_{-1})\boldsymbol{b}_t \\
&= [I - \eta P_1 \boldsymbol{K}_i \boldsymbol{K}_i^T P_1]P_1 \boldsymbol{b}_t - \eta[P_1 \boldsymbol{K}_i \boldsymbol{K}_i^T P_{-1}]P_{-1}\boldsymbol{b}_t
\end{aligned}$$
where $P_1 \boldsymbol{K}_i$ and $P_1 \boldsymbol{b}_t$ are in the same 1 dimensional linear space, thus
$$\begin{aligned}
&P_1 \boldsymbol{K}_i \boldsymbol{K}_i^T P_1 P_1 \boldsymbol{b}_t \\
=&\|P_1 \boldsymbol{K}_i\|_2 \|P_1 \boldsymbol{b}_t\|_2 sign(\langle P_1 \boldsymbol{K}_i, P_1 \boldsymbol{b}_t\rangle)P_1 \boldsymbol{K}_i \\
=&\|P_1 \boldsymbol{K}_i\|_2^2 P_1 \boldsymbol{b}_t \\
\Rightarrow \quad &[I - \eta P_1 \boldsymbol{K}_i \boldsymbol{K}_i^T P_1]P_1 \boldsymbol{b}_t = [1 - \eta\|P_1 \boldsymbol{K}_i\|_2^2]P_1 \boldsymbol{b}_t
\end{aligned}$$
and
$$\begin{aligned}
&\|[P_1 \boldsymbol{K}_i \boldsymbol{K}_i^T P_{-1}]P_{-1}\boldsymbol{b}_t\|_2 \\
\leq &\|P_1 \boldsymbol{K}_i\|_2 \|P_{-1}\boldsymbol{K}_i\|_2 \|P_{-1}\boldsymbol{b}_t\|_2.
\end{aligned}$$
Then
$$E[\|P_1 \boldsymbol{b}_{t+1}\|_2|\boldsymbol{b}_t] \leq E[|1 - \eta\|P_1 \boldsymbol{K}_i\|_2^2|]\|P_1 \boldsymbol{b}_t\|_2 + \eta E[\|P_1 \boldsymbol{K}_i\|_2 \|P_{-1}\boldsymbol{K}_i\|_2]\|P_{-1}\boldsymbol{b}_t\|_2 \tag{22}$$
$$E[\|P_1 \boldsymbol{b}_{t+1}\|_2|\boldsymbol{b}_t] \geq E[|1 - \eta\|P_1 \boldsymbol{K}_i\|_2^2|]\|P_1 \boldsymbol{b}_t\|_2 - \eta E[\|P_1 \boldsymbol{K}_i\|_2 \|P_{-1}\boldsymbol{K}_i\|_2]\|P_{-1}\boldsymbol{b}_t\|_2 \tag{23}$$
where
$$\begin{aligned}
&E[|1 - \eta\|P_1 \boldsymbol{K}_i\|_2^2|] \\
=&\frac{1}{n}\sum_{i=1}^n |1 - \eta\|P_1 \boldsymbol{K}_i\|_2^2| \\
=&\frac{n-1}{n} + \frac{1}{n}|1 - \eta\|P_1 \boldsymbol{K}_1\|_2^2| \\
:=&q_1(\eta)
\end{aligned}$$
and
$$\begin{aligned}
&\eta E[\|P_1 \boldsymbol{K}_i\|_2 \|P_{-1}\boldsymbol{K}_i\|_2] \\
=&\eta\frac{1}{n}\sum_{i=1}^n \|P_1 \boldsymbol{K}_i\|_2 \|P_{-1}\boldsymbol{K}_i\|_2 \\
=&\frac{\eta}{n}\|P_1 \boldsymbol{K}_1\|_2 \|P_{-1}\boldsymbol{K}_1\|_2 \\
\leq &\frac{\eta}{n}(\lambda_1 + \sqrt{n}\tau)c_3(2\lambda_1 + (n-2)\tau)\sqrt{n}\tau \\
\leq &\frac{\eta}{n}c_4\sqrt{n}\tau := \xi(\eta) \tag{24}
\end{aligned}$$

where the first inequality by upper bounds (17) and (18), second inequality by $n\tau \leq \mathcal{O}(1)$. Plug the term (24) into inequalities (22) and (23), take expectation on both sides, we get claims (19) and (20).

For inequality (21), check

$$P_{-1}\boldsymbol{b}_{t+1} = P_{-1}[I - \eta\boldsymbol{K}_i\boldsymbol{K}_i^T]\boldsymbol{b}_t$$
$$= [I - \eta P_{-1}\boldsymbol{K}_i\boldsymbol{K}_i^T P_{-1}]P_{-1}\boldsymbol{b}_t - \eta[P_{-1}\boldsymbol{K}_i\boldsymbol{K}_i^T P_1]P_1\boldsymbol{b}_t$$

Then we have

$$E[\|P_{-1}\boldsymbol{b}_{t+1}\|_2|\boldsymbol{b}_t]$$
$$\leq \frac{1}{n}\sum_{i=1}^{n}\|[I - \eta P_{-1}\boldsymbol{K}_i\boldsymbol{K}_i^T P_{-1}]P_{-1}\boldsymbol{b}_t\|_2 + \eta E[\|P_{-1}\boldsymbol{K}_i\|_2\|P_1\boldsymbol{K}_i\|_2]\|P_1\boldsymbol{b}_t\|_2$$
$$\overset{(24)}{\leq} \frac{1}{n}\sum_{i=1}^{n}\|[I - \eta P_{-1}\boldsymbol{K}_i\boldsymbol{K}_i^T P_{-1}]P_{-1}\boldsymbol{b}_t\|_2 + \xi(\eta)\|P_1\boldsymbol{b}_t\|_2$$

where

$$\frac{1}{n}\sum_{i=1}^{n}\|[I - \eta P_{-1}\boldsymbol{K}_i\boldsymbol{K}_i^T P_{-1}]P_{-1}\boldsymbol{b}_t\|_2$$
$$= \frac{1}{n}\sum_{i=1}^{n}\sqrt{\|[I - \eta P_{-1}\boldsymbol{K}_i\boldsymbol{K}_i^T P_{-1}]P_{-1}\boldsymbol{b}_t\|_2^2}$$
$$= \frac{1}{n}\sum_{i=1}^{n}\sqrt{\|P_{-1}\boldsymbol{b}_t\|_2^2 + \eta^2\|P_{-1}\boldsymbol{K}_i\boldsymbol{K}_i^T P_{-1}P_{-1}\boldsymbol{b}_t\|_2^2 - 2\eta\langle P_{-1}\boldsymbol{b}_t, P_{-1}\boldsymbol{K}_i\boldsymbol{K}_i^T P_{-1}P_{-1}\boldsymbol{b}_t\rangle}$$
$$= \frac{1}{n}\sum_{i=1}^{n}\sqrt{\|P_{-1}\boldsymbol{b}_t\|_2^2 + \eta^2(\boldsymbol{K}_i^T P_{-1}P_{-1}\boldsymbol{b}_t)^2\|P_{-1}\boldsymbol{K}_i\|_2^2 - 2\eta(\boldsymbol{K}_i^T P_{-1}P_{-1}\boldsymbol{b}_t)^2}$$
$$= \frac{1}{n}\sum_{i=1}^{n}\sqrt{1 + \frac{(\boldsymbol{K}_i^T P_{-1}P_{-1}\boldsymbol{b}_t)^2}{\|P_{-1}\boldsymbol{b}_t\|_2^2}(\eta^2\|P_{-1}\boldsymbol{K}_i\|_2^2 - 2\eta)}\|P_{-1}\boldsymbol{b}_t\|_2$$
$$\leq \sqrt{1 + \frac{1}{n}\sum_{i=1}^{n}\frac{(\boldsymbol{K}_i^T P_{-1}P_{-1}\boldsymbol{b}_t)^2}{\|P_{-1}\boldsymbol{b}_t\|_2^2}(\eta^2\|P_{-1}\boldsymbol{K}_i\|_2^2 - 2\eta)}\|P_{-1}\boldsymbol{b}_t\|_2$$

where the last inequality from Jensen's inequality. Now the term

$$\frac{1}{n}\sum_{i=1}^{n}\frac{(\boldsymbol{K}_i^T P_{-1}P_{-1}\boldsymbol{b}_t)^2}{\|P_{-1}\boldsymbol{b}_t\|_2^2}(\eta^2\|P_{-1}\boldsymbol{K}_i\|_2^2 - 2\eta)$$
$$\overset{(11)}{\leq} \frac{1}{n}\sum_{i=1}^{n}\frac{(\boldsymbol{K}_i^T P_{-1}P_{-1}\boldsymbol{b}_t)^2}{\|P_{-1}\boldsymbol{b}_t\|_2^2}\left[\eta^2(\lambda_2^2 + (n-1)\tau^2) - 2\eta\right]$$
$$= \frac{\|KP_{-1}\boldsymbol{b}_t\|_2^2}{n\|P_{-1}\boldsymbol{b}_t\|_2^2}\left[\eta^2(\lambda_2^2 + (n-1)\tau^2) - 2\eta\right]$$

Let

$$q_{-1}(\eta) := \sqrt{1 + \frac{\|KP_{-1}\boldsymbol{b}_t\|_2^2}{n\|P_{-1}\boldsymbol{b}_t\|_2^2}\left[\eta^2(\lambda_2^2 + (n-1)\tau^2) - 2\eta\right]}$$

combine all three inequalities above, take the expectation w.r.t. $\boldsymbol{b}_t$, we have claim (21).

□

**Lemma 26** (Long run behavior of SGD with moderate step size). *Assume $\boldsymbol{b}_0$ is away from 0, $\lambda_n^2 > (2\lambda_1 + (n-2)\tau)n\tau + c_4\sqrt{n}\tau$, $\lambda_2^2 + c_6\sqrt{n}\tau < \lambda_1^2 - c_5\sqrt{n}\tau$ where $c_5, c_6$ are constants such that*

$$c_5 \geq c_3^2[2\lambda_1 + (n-2)\tau]^2\sqrt{n}\tau + c_4$$

$$c_6 \geq \frac{\sqrt{n}\tau - c_4^2 n^{-.5}\tau/[\lambda_n^2 - (2\lambda_1 + (n-2)\tau)n\tau] + \lambda_2^2 c_4/[\lambda_n^2 - [2\lambda_1 + (n-2)\tau]n\tau]}{1 - c_4\sqrt{n}\tau/[\lambda_n^2 - (2\lambda_1 + (n-2)\tau)n\tau]}$$

*Consider first $k_1$ steps of SGD updates with step size $\eta$:*

$$\frac{2}{\lambda_1^2 - c_5\sqrt{n}\tau} < \eta < \frac{2}{\lambda_2^2 + c_6\sqrt{n}\tau}$$

*Fix a $\beta_0 \leq A_0$, then for $0 < \epsilon < 1$ and $0 < \beta < \beta_0$ such that $\sqrt{n}\tau \leq poly(\epsilon\beta)$, there exists $k_1 = \mathcal{O}(\log \frac{1}{\epsilon\beta})$ satisfying:*

- $B_{k_1} \leq \epsilon\beta$

- $A_{k_1} \leq \|\boldsymbol{b}_0\|_2 * \rho_1^{k_1} + \epsilon\beta/2$ *for some $\rho_1 > 1$*

- $A_k > \beta_0$ *for $k = 0, \ldots, k_1$.*

*Proof.* For this choice of $\eta$, denote $q_1 = q_1(\eta), q_{-1} = q_{-1}(\eta), \xi = \xi(\eta)$. By Lemma 21, we have

$$A_k \geq q_1 A_{k-1} - \xi B_{k-1}$$

$$\begin{bmatrix} A_k \\ B_k \end{bmatrix} \leq \begin{bmatrix} q_1 & \xi \\ \xi & q_{-1} \end{bmatrix} \begin{bmatrix} A_{k-1} \\ B_{k-1} \end{bmatrix}$$

Decompose the coefficient matrix as

$$\begin{bmatrix} q_1 & \xi \\ \xi & q_{-1} \end{bmatrix} = \begin{bmatrix} \cos\theta & -\sin\theta \\ \sin\theta & \cos\theta \end{bmatrix} \begin{bmatrix} \rho_1 & 0 \\ 0 & \rho_{-1} \end{bmatrix} \begin{bmatrix} \cos\theta & \sin\theta \\ -\sin\theta & \cos\theta \end{bmatrix}$$

Assume w.l.o.g. that $\sin\theta \geq 0$ ( since otherwise we can take $\theta \to \theta + \pi$), then we have

$$\begin{bmatrix} A_k \\ B_k \end{bmatrix} \leq \begin{bmatrix} q_1 & \xi \\ \xi & q_{-1} \end{bmatrix}^k \begin{bmatrix} A_0 \\ B_0 \end{bmatrix}$$

$$= \begin{bmatrix} \cos\theta & -\sin\theta \\ \sin\theta & \cos\theta \end{bmatrix} \begin{bmatrix} \rho_1^k & 0 \\ 0 & \rho_{-1}^k \end{bmatrix} \begin{bmatrix} \cos\theta & \sin\theta \\ -\sin\theta & \cos\theta \end{bmatrix} \begin{bmatrix} A_0 \\ B_0 \end{bmatrix}$$

$$= \begin{bmatrix} A_0(\rho_1^k \cos^2\theta + \rho_{-1}^k \sin^2\theta) + B_0(\rho_1^k \cos\theta\sin\theta - \rho_{-1}^k \cos\theta\sin\theta) \\ B_0(\rho_{-1}^k \cos^2\theta + \rho_1^k \sin^2\theta) + A_0(\rho_1^k \cos\theta\sin\theta - \rho_{-1}^k \cos\theta\sin\theta) \end{bmatrix}$$

$$= \begin{bmatrix} A_0\rho_1^k + (\rho_1^k - \rho_{-1}^k)\sin\theta(B_0\cos\theta - A_0\sin\theta) \\ B_0\rho_{-1}^k + (\rho_1^k - \rho_{-1}^k)\sin\theta(B_0\sin\theta + A_0\cos\theta) \end{bmatrix}$$

$$\leq \begin{bmatrix} A_0\rho_1^k + |\rho_1^k - \rho_{-1}^k|\sin\theta\sqrt{B_0^2 + A_0^2} \\ B_0\rho_{-1}^k + |\rho_1^k - \rho_{-1}^k|\sin\theta\sqrt{B_0^2 + A_0^2} \end{bmatrix}$$

$$= \begin{bmatrix} A_0\rho_1^k + |\rho_1^k - \rho_{-1}^k|\sin\theta\|\boldsymbol{b}_0\|_2 \\ B_0\rho_{-1}^k + |\rho_1^k - \rho_{-1}^k|\sin\theta\|\boldsymbol{b}_0\|_2 \end{bmatrix}$$

We claim the following holds:

$$0 < \rho_{-1} < 1 < \rho_1 \leq q_1 + \xi \tag{25a}$$

$$\rho_{-1}^{k_1}\|\boldsymbol{b}_0\|_2 \leq \epsilon\beta/2 \tag{25b}$$

$$\rho_1^{k_1}\|\boldsymbol{b}_0\|_2 \sin\theta \leq \epsilon\beta/2 \tag{25c}$$

$$(B_0 + \epsilon\beta_0/2)\xi < (q_1 - 1)\beta_0 \tag{25d}$$

which we check later. Using inequalities (25), we can upper bound $B_{k_1}$ as

$$B_{k_1} \leq B_0\rho_{-1}^{k_1} + (\rho_1^{k_1} - \rho_{-1}^{k_1})\sin\theta\|\boldsymbol{b}_0\|_2$$

$$\leq \|\boldsymbol{b}_0\|_2\rho_{-1}^{k_1} + \rho_1^{k_1}\sin\theta\|\boldsymbol{b}_0\|_2$$

$$\overset{(25b),(25c)}{\leq} \epsilon\beta$$

In addition, for $k = 0, \ldots, k_1$

$$
\begin{aligned}
B_k &\leq B_0 \rho_{-1}^k + (\rho_1^k - \rho_{-1}^k) \sin\theta \|\boldsymbol{b}_0\|_2 \\
&\leq B_0 + \rho_1^{k1} \sin\theta \|\boldsymbol{b}_0\|_2 \\
&\overset{(25c)}{\leq} B_0 + \epsilon\beta/2
\end{aligned}
$$

We now lower bound $A_k$ by mathematical induction. We have $A_0 \geq \beta_0$, assume $A_{k-1} > \beta_0$, then

$$
\begin{aligned}
A_k &\geq q_1 A_{k-1} - \xi B_{k-1} \\
&\geq q_1 \beta_0 - \xi(B_0 + \epsilon\beta/2) \\
&\overset{(25d)}{>} q_1 \beta_0 - (q_1 - 1)\beta_0 = \beta_0
\end{aligned}
$$

For upper bound $A_{k_1}$, check

$$
\begin{aligned}
A_{k_1} &\leq A_0 \rho_1^{k_1} + (\rho_1^{k_1} - \rho_{-1}^{k_1}) \sin\theta \|\boldsymbol{b}_0\|_2 \\
&\leq \|\boldsymbol{b}_0\|_2 \rho_1^{k_1} + \rho_1^{k_1} \sin\theta \|\boldsymbol{b}_0\|_2 \\
&\overset{(25c)}{\leq} \|\boldsymbol{b}_0\|_2 \rho_1^{k_1} + \epsilon\beta/2
\end{aligned}
$$

We have all lemma claims proved. Now it remains to check inequalities (25). First note that our choice of the upper bound on $\eta$ guarantees that $q_{-1} < 1$.

**For inequality** (25a): By Gershgorin circle theorem, it suffices to show $q_{-1} + \xi < 1, q_1 - \xi > 1$, then we have $\rho_1 \geq q_1 - \xi > 1$ and $\rho_{-1} \leq q_{-1} + \xi < 1$. In addition, we need the matrix to be positive definite so that $\rho_{-1} > 0$, we just need $q_1 q_{-1} > \xi^2$ to make the matrix p.d..

$$
\begin{aligned}
&q_{-1} + \xi < 1 \\
\iff &\sqrt{1 + \frac{\|K P_{-1}\boldsymbol{b}_t\|_2^2}{n\|P_{-1}\boldsymbol{b}_t\|_2^2}\left[\eta^2(\lambda_2^2 + (n-1)\tau^2) - 2\eta\right]} < 1 - c_4\eta n^{-1/2}\tau \\
\Longleftarrow &\frac{\|K P_{-1}\boldsymbol{b}_t\|_2^2}{n\|P_{-1}\boldsymbol{b}_t\|_2^2}\left[\eta^2(\lambda_2^2 + (n-1)\tau^2) - 2\eta\right] < c_4^2\eta^2 n^{-1}\tau^2 - 2c_4\eta n^{-1/2}\tau \\
\overset{\text{Lemma } 24}{\Longleftarrow} &2c_4\eta n^{-1/2}\tau \leq c_4^2\eta^2 n^{-1}\tau^2 + \frac{\lambda_n^2 - [2\lambda_1 + (n-2)\tau]n\tau}{n}\left[2\eta - \eta^2(\lambda_2^2 + (n-1)\tau^2)\right] \\
\iff &\frac{\lambda_n^2 - [2\lambda_1 + (n-2)\tau]n\tau}{n}\left[\lambda_2^2 + (n-1)\tau^2\right]\eta - c_4^2 n^{-1}\tau^2\eta \\
&\leq 2\frac{\lambda_n^2 - [2\lambda_1 + (n-2)\tau]n\tau}{n} - 2c_4 n^{-1/2}\tau \\
\Longleftarrow \eta &\leq 2\frac{1 - c_4\sqrt{n}\tau/[\lambda_n^2 - [2\lambda_1 + (n-2)\tau]n\tau]}{\lambda_2^2 + (n-1)\tau^2 - c_4^2\tau^2/[\lambda_n^2 - [2\lambda_1 + (n-2)\tau]n\tau]} \\
\iff \eta &\leq \frac{2}{\lambda_2^2 + \frac{(n-1)\tau^2 - c_4^2\tau^2/[\lambda_n^2 - [2\lambda_1 + (n-2)\tau]n\tau] + \lambda_2^2[c_4\sqrt{n}\tau/[\lambda_n^2 - [2\lambda_1 + (n-2)\tau]n\tau]]}{1 - c_4\sqrt{n}\tau/[\lambda_n^2 - [2\lambda_1 + (n-2)\tau]n\tau]}}}
\end{aligned}
$$

which is true by our choice of $\eta$.

$$
\begin{aligned}
&q_1 - \xi > 1 \\
\iff &\frac{n-1}{n} + \frac{1}{n}|1 - \eta\|P_1\boldsymbol{K}_1\|_2^2| - c_4\eta n^{-1/2}\tau > 1 \\
\iff &1 + c_4\eta\sqrt{n}\tau < |1 - \eta\|P_1\boldsymbol{K}_1\|_2^2| \\
\Longleftarrow &\eta\|P_1\boldsymbol{K}_1\|_2^2 - 1 > 1 + c_4\eta\sqrt{n}\tau \\
\Longleftarrow &\eta > \frac{2}{\|P_1\boldsymbol{K}_1\|_2^2 - c_4\sqrt{n}\tau} \\
\overset{(18)}{\Longleftarrow} &\eta > \frac{2}{\lambda_1^2 - c_3^2[2\lambda_1 + (n-2)\tau]^2 n\tau^2 - c_4\sqrt{n}\tau}
\end{aligned}
$$

which is true by our lower bound on $\eta$.

$$
\begin{aligned}
& q_1 q_{-1} > \xi^2 \\
\Longleftarrow\; & q_{-1}^2 \geq \xi \\
\Longleftrightarrow\; & 1 + \frac{\|KP_{-1}\boldsymbol{b}_t\|_2^2}{n\|P_{-1}\boldsymbol{b}_t\|_2^2}\left[\eta^2(\lambda_2^2 + (n-1)\tau^2) - 2\eta\right] > c_4 \eta n^{-1/2}\tau \\
\Longleftarrow\; & 1 - 2\eta\frac{\lambda_2^2 + (2\lambda_1 + (n-1)\tau)n\tau}{n} > c_4\eta n^{-1/2}\tau \\
\Longleftrightarrow\; & \eta < \frac{n}{2\lambda_2^2 + 2(2\lambda_1 + (n-1)\tau)n\tau + c_4\sqrt{n}\tau}
\end{aligned}
$$

which is true.

**For inequality** (25b): It suffices to take

$$
k_1 = \frac{\log(\epsilon\beta/(2\|\boldsymbol{b}_0\|_2))}{\log(\rho_{-1})} = \mathcal{O}(\log\frac{1}{\epsilon\beta})
$$

**For inequality** (25c): We just need to show $\sin\theta < (\rho_{-1}/\rho_1)^{k_1}$. Calculate that $\xi/(q_1 - q_{-1}) = \frac{\cos\theta\sin\theta}{\cos^2\theta - \sin^2\theta}$, then

$$
\begin{aligned}
& \sin\theta < (\rho_{-1}/\rho_1)^{k_1} \\
\Longleftarrow\; & \xi/(q_1 - q_{-1}) < 0.9(\rho_{-1}/\rho_1)^{k_1} \\
\Longleftarrow\; & \xi < 0.9(q_1 - q_{-1})(\frac{q_{-1} - \xi}{q_1 + \xi})^{k_1} \\
\Longleftarrow\; & \xi < 0.9(q_1 - q_{-1})(\epsilon\beta/(2\|\boldsymbol{b}_0\|_2))^{1 - \frac{\log(q_1 + \xi)}{\log(q_{-1} - \xi)}} \\
\Longleftarrow\; & \xi < (q_1 - 1)poly(\epsilon\beta) \\
\Longleftarrow\; & \sqrt{n}\tau \leq poly(\epsilon\beta)
\end{aligned}
$$

**For inequality** (25d): Suffice to show

$$
\begin{aligned}
& \xi < \frac{(q_1 - 1)\beta_0}{B_0 + \epsilon\beta_0/2} \\
\Longleftarrow\; & \sqrt{n}\tau \leq \mathcal{O}(1)
\end{aligned}
$$

$\square$

**Lemma 27** (Long run behavior of SGD with small step size). *Under the same notations and assumptions as Lemma* 26 *unless otherwise specified. Consider another $k_2 - k_1$ steps of SGD update with step size*

$$
\eta' < \frac{1}{\lambda_1^2 + c_7\sqrt{n}\tau}
$$

*where the constant $c_7 \geq \sqrt{n}\tau + c_4$. Then we have for $k > k_1$:*

- $B_k \leq \epsilon\beta$

- $A_k \leq \begin{cases} qA_{k-1} & , A_{k-1} > \beta \\ \beta & , A_{k-1} < \beta \end{cases}$

*where $q := q_1(\eta') + \xi(\eta')\epsilon < 1$.*

*Proof.* Denote $q_1' = q_1(\eta'), q_{-1}' = q_{-1}(\eta'), \xi' = \xi(\eta')$, then $q = q_1' + \xi'\epsilon$, denote $B = \|\boldsymbol{b}_0\|_2 * \rho_1^{k_1} + \epsilon\beta/2$. We have by proof of Lemma 26 that $q_{-1}' + \xi < 1$. We claim the following holds:

$$
\begin{aligned}
q &< 1 & \text{(26a)} \\
\xi'B &\leq (1 - q_1')\epsilon\beta & \text{(26b)}
\end{aligned}
$$

**Check inequality** (26a):

$$
\begin{aligned}
&q_1' + \xi'\epsilon < 1 \\
\Longleftarrow\ &\frac{n-1}{n} + \frac{1}{n}|1 - \eta'\|P_1 \boldsymbol{K}_1\|_2^2| + c_4 \eta' n^{-1/2}\tau < 1 \\
\Longleftrightarrow\ &|1 - \eta'\|P_1 \boldsymbol{K}_1\|_2^2| < 1 - c_4 \eta' \sqrt{n}\tau \\
\Longleftrightarrow\ &c_4 \eta' \sqrt{n}\tau < \eta'\|P_1 \boldsymbol{K}_1\|_2^2 < 2 - c_4 \eta' \sqrt{n}\tau \\
\Longleftrightarrow\ &\left\{
\begin{array}{l}
c_4 \sqrt{n}\tau < \|P_1 \boldsymbol{K}_1\|_2^2 \\
\eta' < \frac{2}{\|P_1 \boldsymbol{K}_1\|_2^2 + c_4 \sqrt{n}\tau}
\end{array}
\right. \\
\Longleftarrow\ &\left\{
\begin{array}{l}
\sqrt{n}\tau < \mathcal{O}(1) \\
\eta' < \frac{2}{\lambda_1^2 + n\tau^2 + c_4 \sqrt{n}\tau}
\end{array}
\right.
\end{aligned}
$$

which are true by assumption.

**Check inequality** (26b):

$$
\begin{aligned}
&\xi' B \le (1 - q_1')\epsilon\beta \\
\Longleftrightarrow\ &c_4 \eta' n^{-1/2}\tau(\|\boldsymbol{b}_0\|_2 * \rho_1^{k_1} + \epsilon\beta/2) \le (1 - \frac{n-1}{n} - \frac{1}{n}(1 - \eta'\|P_1 \boldsymbol{K}_1\|_2^2))\epsilon\beta \\
\Longleftrightarrow\ &c_4 \eta' \sqrt{n}\tau(\|\boldsymbol{b}_0\|_2 * \exp(k_1)^{\log \rho_1} + \epsilon\beta/2) \le \eta'\|P_1 \boldsymbol{K}_1\|_2^2 \epsilon\beta \\
\Longleftarrow\ &\sqrt{n}\tau \le poly(\epsilon\beta).
\end{aligned}
$$

With (26) we can prove the lemma by mathematical induction. Suppose $B_{k-1} \le \epsilon\beta$ and $A_{k-1} \le A_{k_1} \le B$, then check

$$
\begin{aligned}
B_k &\le \xi' A_{k-1} + q_{-1}' B_{k-1} \\
&\le \xi' B + q_{-1}' \epsilon\beta \\
&\overset{(26b)}{\le} \epsilon\beta
\end{aligned}
$$

and

$$
\begin{aligned}
A_k &\le q_1' A_{k-1} + \xi' B_{k-1} \\
&\le q_1' A_{k-1} + \xi' \epsilon\beta \\
&\le (q_1' + \xi'\epsilon)\max\{A_{k-1}, \beta\} \\
&\le \left\{
\begin{array}{ll}
q A_{k-1} & , A_{k-1} > \beta \\
q\beta < \beta & , A_{k-1} < \beta
\end{array}
\right. .
\end{aligned}
$$

$\square$

We recap Theorem 5 using our notations in previous lemmas as follows:

**Theorem 28** (Directional bias of the two-stage SGD)**.** *Use the two stage SGD scheme as defined in Lemma 26 and 27. Assume $n\tau < poly(\epsilon)$, then there exists $k_1 = \mathcal{O}(\log \frac{1}{\epsilon})$ and $k_2$ such that*

$$
(1 - 2\epsilon)\gamma_1 \le \frac{E[\|K\boldsymbol{b}_{k_2}\|_2]}{E[\|\boldsymbol{b}_{k_2}\|_2]} \le \gamma_1
$$

*where $\gamma_1$ is the largest eigenvalue of $K$.*

*Proof.* In Lemma 26 let $\beta = \beta_0$, then for $k_1 = \mathcal{O}(\log \frac{1}{\epsilon})$ we have $B_{k_1} \le \epsilon\beta_0$. For the 2nd stage, by Lemma 27 we can early stop at $k_2$ such that $A_{k_2} \ge \beta_0$ and $A_{k_2+1} < \beta_0$. We then have

$$
B_{k_2} \le \epsilon\beta_0 \le \epsilon A_{k_2}
$$

Then we check

$$\frac{E\|K\boldsymbol{b}_{k_2}\|_2}{E\|\boldsymbol{b}_{k_2}\|_2}$$

$$=\frac{E\sqrt{\|KP_{-1}\boldsymbol{b}_{k_2}\|_2^2+\|KP_1\boldsymbol{b}_{k_2}\|_2^2+2\langle\boldsymbol{K}_1^TP_{-1}\boldsymbol{b}_{k_2},\boldsymbol{K}_1^TP_1\boldsymbol{b}_{k_2}\rangle}}{E\|\boldsymbol{b}_{k_2}\|_2}$$

$$\geq\frac{E\sqrt{\|KP_1\boldsymbol{b}_{k_2}\|_2^2-2\|P_{-1}\boldsymbol{K}_1\|_2\|P_1\boldsymbol{K}_1\|_2\|\boldsymbol{b}_{k_2}\|_2^2}}{E\|\boldsymbol{b}_{k_2}\|_2}$$

$$\geq\frac{E\sqrt{\|KP_1\boldsymbol{b}_{k_2}\|_2^2}-E\sqrt{2\|P_{-1}\boldsymbol{K}_1\|_2\|P_1\boldsymbol{K}_1\|_2\|\boldsymbol{b}_{k_2}\|_2^2}}{E\|\boldsymbol{b}_{k_2}\|_2}$$

$$=\frac{E\|\boldsymbol{K}_1^TP_1\boldsymbol{b}_{k_2}\|_2-\sqrt{2\|P_{-1}\boldsymbol{K}_1\|_2\|P_1\boldsymbol{K}_1\|_2}E\|\boldsymbol{b}_{k_2}\|_2}{E\|\boldsymbol{b}_{k_2}\|_2}$$

$$\overset{(17),(18)}{\geq}\sqrt{\lambda_1^2-c_3^2[2\lambda_1+(n-2)\tau]^2n\tau^2}\frac{E\|P_1\boldsymbol{b}_{k_2}\|_2}{E\|P_1\boldsymbol{b}_{k_2}\|_2+E\|P_{-1}\boldsymbol{b}_{k_2}\|_2}$$

$$\quad-\sqrt{2(\lambda_1+\sqrt{n}\tau)(c_3(2\lambda_1+(n-2)\tau)\sqrt{n}\tau)}$$

$$\geq\sqrt{\lambda_1^2-c_3^2[2\lambda_1+(n-2)\tau]^2n\tau^2}\frac{\beta_0}{\epsilon\beta_0+\beta_0}-\sqrt{2(\lambda_1+\sqrt{n}\tau)(c_3(2\lambda_1+(n-2)\tau)\sqrt{n}\tau)}$$

$$\overset{(13)}{\geq}(\gamma_1-n\tau-c_3(2\lambda_1+(n-2)\tau)\sqrt{n}\tau)(1-\epsilon)-\sqrt{2(\lambda_1+\sqrt{n}\tau)(c_3(2\lambda_1+(n-2)\tau)\sqrt{n}\tau)}$$

$$\geq\gamma_1(1-\epsilon)-\gamma_1\epsilon\quad(\text{By }n\tau<poly(\epsilon))$$

$$=\gamma_1(1-2\epsilon)$$

And the upper bound in the theorem is by definition of $\gamma_1$. $\qquad\square$

## F   DIRECTIONAL BIAS OF GD WITH MODERATE OR SMALL STEP SIZE

This section includes the proof of Theorem 7. We first rewrite the GD updates as linear combination of eigenvectors. Then the theorem is proved using the transformed variables and finally transformed back to original parameters.

**The directional bias of GD does not require diagonal dominant gram matrix.**

**Reloading notations** Denote the eigen decomposition of $K$:

$$K=G\Gamma G^T,\Gamma=diag(\gamma_1,\ldots,\gamma_n),G=[\boldsymbol{g}_1,\ldots,\boldsymbol{g}_n]$$

where the eigenvectors $\boldsymbol{g}_i$'s are orthogonal. The GD update as

$$\boldsymbol{\alpha}_{t+1}=\boldsymbol{\alpha}_t-\frac{\eta}{n}K(K\boldsymbol{\alpha}_t-\boldsymbol{y})$$

Denote $\boldsymbol{w}_t:=G^T(\boldsymbol{\alpha}_t-\hat{\boldsymbol{\alpha}})$, we can rewrite GD update in $\boldsymbol{w}_t$:

$$\boldsymbol{w}_{t+1}=\boldsymbol{w}_t-\frac{\eta}{n}\Gamma^2\boldsymbol{w}_t=(I-\frac{\eta}{n}\Gamma^2)\boldsymbol{w}_t$$

We recap Theorem 7 to make reading easier as follows:

**Theorem 29** (Direction bias of GD). *Assume $\boldsymbol{\alpha}_0$ is away from $0$, $\lambda_n+2n\tau<\lambda_{n-1}$, GD with step size:*

$$\eta<\frac{n}{(\lambda_1+n\tau)^2}$$

*For a small $\epsilon>0$, take $k=\mathcal{O}(\log\frac{1}{\epsilon})$, we have*

$$\gamma_n\leq\frac{\|K(\boldsymbol{\alpha}_k-\hat{\boldsymbol{\alpha}})\|_2}{\|\boldsymbol{\alpha}_k-\hat{\boldsymbol{\alpha}}\|_2}\leq\sqrt{1+\epsilon}\gamma_n$$

*Proof.* For $i = 1, \ldots, n$, we have

$$w_k^{(i)} = (1 - \eta\gamma_i^2/n)^k w_0^{(i)}$$

Denote $q_i = 1 - \eta\gamma_i^2/n$, then $0 < q_1 \leq \ldots \leq q_n < 1$ since

$$0 < \eta < \frac{n}{(\lambda_1 + n\tau)^2} \overset{(13)}{\leq} \frac{n}{\gamma_1^2} \leq \frac{n}{\gamma_i^2}$$

Since $\lambda_n + n\tau < \lambda_{n-1} - n\tau$, we have $\gamma_n < \gamma_{n-1}$ by lemma 20, it follows that $q_n > q_{n-1}$. Denote $q = q_{n-1}/q_n < 1$, then

$$\frac{\sum_{i=1}^{n-1}(w_k^{(i)})^2}{(w_k^{(n)})^2}$$

$$= \frac{\sum_{i=1}^{n-1} q_i^{2k}(w_0^{(i)})^2}{q_n^{2k}(w_0^{(n)})^2}$$

$$\leq \frac{\sum_{i=1}^{n-1} q_{n-1}^{2k}(w_0^{(i)})^2}{q_n^{2k}(w_0^{(n)})^2}$$

$$= q^{2k}\frac{\sum_{i=1}^{n-1}(w_0^{(i)})^2}{(w_0^{(n)})^2}$$

Let $q^{2k} \leq \frac{\gamma_n^2 \epsilon (w_0^{(n)})^2}{\gamma_1^2 \sum_{i=1}^{n-1}(w_0^{(i)})^2} \iff k \geq \frac{1}{2}\frac{\log\frac{\gamma_n^2 \epsilon (w_0^{(n)})^2}{\gamma_1^2 \sum_{i=1}^{n-1}(w_0^{(i)})^2}}{\log q} = \mathcal{O}(\log\frac{1}{\epsilon})$, we have

$$\frac{\sum_{i=1}^{n-1}(w_k^{(i)})^2}{(w_k^{(n)})^2} \leq \frac{\gamma_n^2 \epsilon}{\gamma_1^2}$$

Thus

$$\frac{\|K(\boldsymbol{\alpha}_k - \hat{\boldsymbol{\alpha}})\|_2^2}{\|\boldsymbol{\alpha}_k - \hat{\boldsymbol{\alpha}}\|_2^2} = \frac{\|\Gamma \boldsymbol{w}_k\|_2^2}{\|\boldsymbol{w}_k\|_2^2}$$

$$= \frac{\sum_{i=1}^{n}(w_k^{(i)})^2\gamma_i^2}{\sum_{i=1}^{n}(w_k^{(i)})^2}$$

$$= \frac{(w_k^{(n)})^2\gamma_n^2}{\sum_{i=1}^{n}(w_k^{(i)})^2} + \frac{\sum_{i=1}^{n-1}(w_k^{(i)})^2\gamma_i^2}{\sum_{i=1}^{n}(w_k^{(i)})^2}$$

$$\leq \gamma_n^2 + \frac{\sum_{i=1}^{n-1}(w_k^{(i)})^2}{\sum_{i=1}^{n}(w_k^{(i)})^2}\gamma_1^2$$

$$\leq \gamma_n^2 + \gamma_1^2\frac{\gamma_n^2}{\gamma_1^2}\epsilon = \gamma_n^2(1 + \epsilon)$$

thus

$$\frac{\|K(\boldsymbol{\alpha}_k - \hat{\boldsymbol{\alpha}})\|_2}{\|\boldsymbol{\alpha}_k - \hat{\boldsymbol{\alpha}}\|_2} \leq \sqrt{\gamma_n^2(1 + \epsilon)}$$

The lower bound of the theorem holds by definition of $\gamma_n$. □

## G   EFFECT OF DIRECTIONAL BIAS

In this section, we provide the proof for theorems in Section 4.2. There are two theorems there, so we split this section into two subsections. Subsection G.1 proves Theorem 9, for a general problem setting of squared error minimization, it provides a straightforward understanding for why directional bias towards the largest eigenvalue of the Hessian is good for generalization. Section G.2 proves Theorem 11 by giving concrete generalization bounds of SGD and GD estimators in kernel regression.

## G.1 PROOF OF THEOREM 9

Denote $v = w - w^*$, rewrite the objective function as

$$\min_{v} \quad \|v\|_2^2$$
$$s.t. \quad \|Av\|_2^2 = a$$

Denote the eigen decomposition of $A^T A = Q\Gamma Q^T$ where $Q = [q_1, \ldots, q_n]$, $QQ^T = Q^T Q = I$ and $\Gamma = diag([\rho_1, \ldots, \rho_n])$, $\rho_1 \geq \ldots \geq \rho_n \geq 0$. Then

$$\|Av\|_2^2 = \sum_{i=1}^{n} \rho_i (q_i^T v)^2$$

So

$$\|Av\|_2^2 \leq \rho_1 [\sum_{i=1}^{n} (q_i^T v)^2] = \rho_1 v^T QQ^T v = \rho_1 \|v\|_2^2$$

The equality is achieved when $v$ is in the direction of $q_1$, and $\rho_1 = \|A^T A\|_2$. Take $L(w) = \|Av\|_2^2 = a$ then the theorem holds.

## G.2 PROOF OF THEOREM 11

**Calculate $\Delta_a^*$:** Denote $f^* = \hat{\alpha}^T K(\cdot, X) + \tilde{f}$, then we have for a $f \in \mathcal{H}_s$, $f = \alpha^T K(\cdot, X)$, let $b = \hat{\alpha} - \alpha$, then

$$\|f^* - f\|_{\mathcal{H}}^2$$
$$= \|b^T K(\cdot, X) + \tilde{f}\|_{\mathcal{H}}^2$$
$$= \|b^T K(\cdot, X)\|_{\mathcal{H}}^2 + \|\tilde{f}\|_{\mathcal{H}}^2 + 2\langle b^T K(\cdot, X), \tilde{f} \rangle_{\mathcal{H}}$$

where we can check

$$\langle b^T K(\cdot, X), \tilde{f} \rangle_{\mathcal{H}} = \sum_{i=1} b_i \langle K(\cdot, x_i), f^* - \hat{\alpha}^T K(\cdot, X) \rangle_{\mathcal{H}}$$
$$= \sum_{i=1} b_i [\langle K(\cdot, x_i), f^* \rangle_{\mathcal{H}} - \langle K(\cdot, x_i), \hat{\alpha}^T K(\cdot, X) \rangle_{\mathcal{H}}]$$
$$= \sum_{i=1} b_i [f^*(x_i) - \hat{\alpha}^T K(x_i, X)] (\text{By reproducing property})$$
$$= \sum_{i=1} b_i [y_i - y_i] = 0$$

And we further calculate that

$$\|b^T K(\cdot, X)\|_{\mathcal{H}}^2 = \langle \sum_{i=1}^{n} b_i K(\cdot, x_i), \sum_{j=1}^{n} b_j K(\cdot, x_j) \rangle_{\mathcal{H}}$$
$$= \sum_{i,j=1}^{n} b_i b_j \langle K(\cdot, x_i), K(\cdot, x_j) \rangle$$
$$= \sum_{i,j=1}^{n} b_i b_j K(x_i, x_j) = b^T K b$$

That is,

$$L_D(f) = b^T K b + \|\tilde{f}\|_{\mathcal{H}}^2$$

and

$$\inf_{f \in \mathcal{H}_s} L_D(f) = \|\tilde{f}\|_{\mathcal{H}}^2$$

It follows that

$$\Delta(f) = L_D(f) - \inf_{f \in \mathcal{H}_s} L_D(f) = \boldsymbol{b}^T K \boldsymbol{b}$$

We claim that

$$\Delta_a^* = \min_{\boldsymbol{b}: \frac{1}{2n} \|K\boldsymbol{b}\|_2^2 = a} \boldsymbol{b}^T K \boldsymbol{b} = \frac{1}{\gamma_1} \|K\boldsymbol{b}\|_2^2 = 2na/\gamma_1$$

where the equality is obtained when $\boldsymbol{b}$ is in the direction of the largest eigenvector of $K$. To see this, we check $\|K\boldsymbol{b}\|_2^2 \le \gamma_1 \boldsymbol{b}^T K \boldsymbol{b}$. Recall the eigendecomposition of $K = G\Gamma G^T$ where $G = [\boldsymbol{g}_1, \dots, \boldsymbol{g}_n]$ has orthogonal columns and $\Gamma = diag(\gamma_1, \dots, \gamma_n)$. Then

$$\|K\boldsymbol{b}\|_2^2 = \sum_{i=1}^n \gamma_i^2 (\boldsymbol{g}_i^T \boldsymbol{b})^2$$

and

$$\boldsymbol{b}^T K \boldsymbol{b} = \sum_{i=1}^n \gamma_i (\boldsymbol{g}_i^T \boldsymbol{b})^2$$

So we have

$$\|K\boldsymbol{b}\|_2^2 \le \gamma_1 [\sum_{i=1}^n \gamma_i (\boldsymbol{g}_i^T \boldsymbol{b})^2] = \gamma_1 \boldsymbol{b}^T K \boldsymbol{b}$$

This finishes our claim on $\Delta_a^*$.

**SGD output:** By Theorem 5, the SGD output has

$$(1 - 2\epsilon)\gamma_1 E[\|\boldsymbol{b}_{k_2}\|_2] \le E[\|K\boldsymbol{b}_{k_2}\|_2]$$

Thus

$$\begin{aligned}
E[\Delta^{1/2}(f^{SGD})] &= E\left[\sqrt{\sum_{i=1}^n \gamma_i (\boldsymbol{g}_i^T \boldsymbol{b}^{SGD})^2}\right] \\
&\le \sqrt{\gamma_1} E[(\sum_{i=1}^n (\boldsymbol{g}_i^T \boldsymbol{b}^{SGD})^2)^{1/2}] \\
&= \sqrt{\gamma_1} E\|\boldsymbol{b}^{SGD}\|_2 \\
&\le \sqrt{\gamma_1} E[\|K\boldsymbol{b}^{SGD}\|_2]/[(1 - 2\epsilon)\gamma_1] \\
&= \sqrt{2na/\gamma_1}/(1 - 2\epsilon) \\
&= \frac{1}{1 - 2\epsilon}(\Delta_a^*)^{1/2} \\
&< (1 + 4\epsilon)(\Delta_a^*)^{1/2}
\end{aligned}$$

last inequality by let $\epsilon < 1/4$.

**GD output:** By Theorem 7, the GD output has

$$\frac{\|K\boldsymbol{b}^{GD}\|_2^2}{\|\boldsymbol{b}^{GD}\|_2^2} \le (1 + \epsilon')\gamma_n^2$$

Thus

$$
\begin{aligned}
\Delta(f^{GD}) &= \sum_{i=1}^{n} \gamma_i (\boldsymbol{g_i}^T \boldsymbol{b}^{GD})^2 \\
&\geq \gamma_n \sum_{i=1}^{n} (\boldsymbol{g_i}^T \boldsymbol{b}^{GD})^2 \\
&= \gamma_n \|\boldsymbol{b}^{GD}\|_2^2 \\
&\geq \gamma_n \|K\boldsymbol{b}^{GD}\|_2^2 / [(1+\epsilon')\gamma_n^2] \\
&= 2na / [(1+\epsilon')\gamma_n] \\
&= \frac{\gamma_1}{(1+\epsilon')\gamma_n} \Delta_a^* \\
&> \frac{\gamma_1}{\gamma_n}(1-\epsilon')\Delta_a^* \\
&:= M\Delta_a^*
\end{aligned}
$$

where $M > 1$ by taking $\epsilon' < 1 - \gamma_n/\gamma_1$.

## H  EXPERIMENTS

We list the implementation details of the experiments at the end of Section 4 and include more experiment results. For better presenting, we split into two subsections: Subsection H.1 includes the details of simulation; Subsection H.2 is about the NN experiment on FashionMNIST, including the data description, network structure, and algorithm details, also there are more experiment results in Subsection H.2 that are not listed in Section 4 due to page limit.

### H.1  SIMULATION

This subsection is corresponding to Figure 1.

**Data Generation**. The training data is simulated as follows: Set $n = 10, p = 100$, simulate $X_{n \times p}$ where elements of $X$ are i.i.d. $N(0,1)$; denote $i$th row of $X$ as $\boldsymbol{x}_i$, normalize $\boldsymbol{x}_i$ such that it has squared $\ell_2$ norm in $[.49, 1]$; set $y_i = \sum_{j=1}^{p} \sin(x_{i,j}) + \epsilon_i$ where $\epsilon_i \overset{i.i.d.}{\sim} N(0, .01)$. The testing data is simulated in exactly the same way, except that we only simulate $n = 5$ testing data.

**Kernel Function**. We set the kernel function to be the polynomial kernel

$$
K(\boldsymbol{x}_1, \boldsymbol{x}_2) = (\langle \boldsymbol{x}_1, \boldsymbol{x}_2 \rangle + .01)^2
$$

**SGD and GD implementation**. Both SGD and GD is run for small and moderate step sizes. The moderate step size scheme for SGD is: $\eta_1 = .1$ for the first 50 steps, and $\eta_2 = .01$ for the next 1000 steps; for GD is: $\eta_1 = .5$ for the first 50 steps, and $\eta_2 = .05$ for the next 1000 steps. The small step size scheme for SGD is $\eta = 0.01$ for 1050 steps; for GD is $\eta = 0.05$ for 1050 steps. Note that the step size for SGD is a fraction of that for GD, this matches our Theorem 5 and 7 that the step size of GD is of magnitude $n/2$ times that of SGD.

### H.2  NEURAL NETWORK ON FASHIONMNIST

This subsection is corresponding to Figure 2.

**Dataset**. The original FashionMNIST consist of $60,000$ training data and $10,000$ testing data. We randomly sample $1,500$ data from original training data for training, and use all $10,000$ original testing data for testing. All data entries are normalized to $[0, 1]$.

**Network structure**. we use a 6-layer ResNet-like (He et al., 2016) Neural Network, and the structure is as follows

$$
\text{Input} \Rightarrow 7 \times 7 \text{ Conv} \Rightarrow \text{BatchNorm} \Rightarrow \text{ReLU} \Rightarrow 3 \times 3 \text{ MaxPool}
$$
$$
\Rightarrow \text{ResBlock1} \Rightarrow \text{ResBlock2} \Rightarrow \text{Global AvePool} \Rightarrow \text{FC} \Rightarrow \text{output}
$$

The Residual Blocks are as Figure 7.6.3 in Zhang et al. (2021) (without $1 \times 1$ convolution). Note that each residual block contains two $3 \times 3$ convolutional layers, thus total number of layers is as stated.

**Algorithm**. We minimize the Cross Entropy Loss objective $L(\mathbf{w}) = \frac{1}{n} \sum_{i=1}^{n} l_i(\mathbf{w})$, where $l_i(\mathbf{w})$ is the loss function at $i$th sample. One step SGD is as follows:

$$\boldsymbol{w}_{t+1} = \boldsymbol{w}_t - \eta_t \frac{1}{|I|} \sum_{i \in I} \nabla l_i(\boldsymbol{w}_t)$$

where $I$ is a randomly sampled subset of $\{1, \ldots, n\}$ (uniform random sample without replacement). We choose the batch size $|I|$ to be 25.

One step GD is as follows:

$$\boldsymbol{w}_{t+1} = \boldsymbol{w}_t - \eta_t \nabla L(\boldsymbol{w}_t)$$

Both SGD and GD are run using two settings of step sizes $\eta_t$. The moderate step size setting is as follows:

$$\eta_t = \begin{cases} 0.2, & t = 1, \ldots, 5000 \\ 0.02, & t = 5001, \ldots, 20000 \end{cases}$$

And the small step size setting has $\eta_t = 0.02, t = 1, \ldots, 20000$.

**Comparison of convergence direction**. Since the loss surface is nonconvex and the Hessian varies, we follow Wu et al. (2021) to measure the convergence direction by Relative Rayleigh Quotient(RRQ), which normalizes the Rayleigh Quotient by the maximum eigenvalue of the Hessian as follows

$$RRQ(\boldsymbol{w}) = \frac{\frac{\nabla L(\boldsymbol{w})^\top}{\|\nabla L(\boldsymbol{w})\|_2} \cdot \nabla^2 L(\boldsymbol{w}) \cdot \frac{\nabla L(\boldsymbol{w})}{\|\nabla L(\boldsymbol{w})\|_2}}{\|\nabla^2 L(\boldsymbol{w})\|_2}$$

where $L(w)$ is the loss function on the whole training set. A high RRQ indicates that the convergence direction of $w$ is close to a larger eigenvector of the Hessian.

**Comparison of test accuracy**. We set 20 different random seeds. For each random seed, we run: SGD with moderate step size, GD with moderate step size, SGD with small step size, GD with small step size. For each algorithm, we evaluate its test accuracy once every 500 steps, and use the average of the last 5 values as its test accuracy. We list the test accuracy in Table 1.

| Experiment | #1 | #2 | #3 | #4 | #5 | #6 | #7 | #8 | #9 | #10 |
|---|---|---|---|---|---|---|---|---|---|---|
| SGD + moderate LR | 83.69 | 82.95 | 82.37 | 82.05 | 83.4 | 83.16 | 83.72 | 83.29 | 83.28 | 83.23 |
| GD + moderate LR | 80.93 | 80.79 | 80.79 | 81.80 | 81.68 | 81.12 | 82.43 | 81.63 | 80.94 | 81.54 |
| SGD + small LR | 82.00 | 81.72 | 81.34 | 81.92 | 82.63 | 82.67 | 82.99 | 82.22 | 80.78 | 82.10 |
| GD + small LR | 78.88 | 78.71 | 78.49 | 79.3 | 80.45 | 79.78 | 80.15 | 79.66 | 79.54 | 79.68 |
| Experiment | #11 | #12 | #13 | #14 | #15 | #16 | #17 | #18 | #19 | #20 |
| SGD + moderate LR | 83.12 | 82.92 | 83.58 | 83.47 | 82.35 | 83.57 | 83.59 | 82.43 | 84.21 | 83.12 |
| GD + moderate LR | 82.41 | 81.56 | 81.42 | 80.86 | 81.23 | 81.25 | 81.82 | 80.42 | 81.80 | 82.12 |
| SGD + small LR | 82.62 | 80.66 | 82.01 | 81.01 | 81.32 | 81.66 | 82.12 | 80.78 | 82.28 | 82.48 |
| GD + small LR | 80.08 | 78.29 | 79.93 | 79.36 | 78.9 | 79.69 | 80.2 | 79.62 | 79.98 | 79.69 |

Table 1: Test Accuracy

We also use one-side Wilcoxon signed-rank test to check if the test accuracy of different algorithm are significantly different, the result is in Table 2. All the p-values are significant at $0.01$ level, so we reject the null hypothesis and conclude that the SGD with moderate step size has test accuracy significantly higher than all other algorithms.

| Null Hypothesis on Test Accuracy | p-value |
|---|---|
| SGD + moderate LR $\leq$ GD + moderate LR | $9.54 \times 10^{-7}$ |
| SGD + moderate LR $\leq$ SGD + small LR | $9.54 \times 10^{-7}$ |
| SGD + moderate LR $\leq$ GD + small LR | $9.54 \times 10^{-7}$ |

Table 2: Wilcoxon signed-rank test result

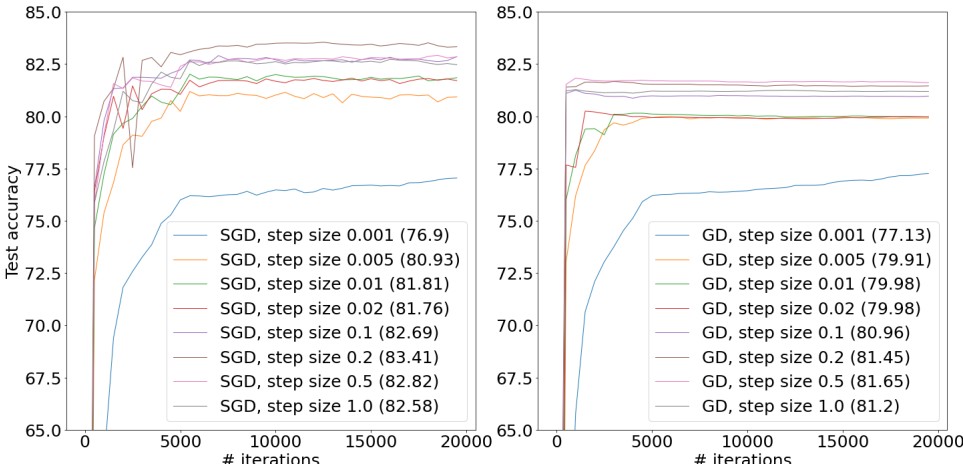

Figure 3: Use more step sizes in SGD/GD. The test accuracy is evaluated once every 500 iterations, and inside the bracket is the average of the last 5 test accuracy values.

**Additional experiments**. We conduct more experiments using different step sizes. The initial step size is taken in $\{1, 0.5, 0.2, 0.1, 0.02, 0.01, 0.005, 0.001\}$, and the step size is divided by a factor of 10 after 5000 steps. The test accuracy is in Figure 3, where we see that SGD with step size 0.2 has the best test accuracy, and GD with step size 0.5 performs better than GD with any other step sizes, but is still worse than the best SGD.

