# OpenReview forum: "Directional Bias Helps Stochastic Gradient Descent to Generalize in Nonparametric Model"
_ICLR.cc/2022/Conference — ICLR 2022 Submitted_

### Official Review · Reviewer_EcVL · 2021-11-02

**Correctness:** 4
**Technical Novelty And Significance:** 2
**Empirical Novelty And Significance:** 2
**Recommendation:** 5
**Confidence:** 5

**Main Review:**

# Pros:
+ The paper is very well written and easy to follow. Related works are well discussed.
+ Prior work only shows the directional bias of SGD for linear regression. It is good to learn that the directional bias of SGD also works for kernel regression.

# Cons:
- Most of the proof techniques are from (Wu et al. 2021). Given prior results for linear regression, an extension to kernel regression seems a bit incremental to justify a conference paper.

**Summary Of The Paper:**

This paper studies directional bias of SGD vs. GD in the setting of kernel regression. The presented results well-recover those for linear regression shown by prior work (Wu et al. 2021). Both theory and experiments are presented.

**Summary Of The Review:**

I cannot recommend an acceptance to this paper as most of the results are more or less incremental given prior work.

---

### Official Review · Reviewer_aXTx · 2021-11-03

**Correctness:** 4
**Technical Novelty And Significance:** 2
**Empirical Novelty And Significance:** 2
**Recommendation:** 3
**Confidence:** 4

**Main Review:**


Basically, this paper is well organized and the results are clearly presented. However, my major concern is regarding the novelty of this paper, given the closely related prior work [Wu et al., 2021].


**1.** It seems that given Assumption 1 and Proposition 14, the model is pretty similar to the linear setting considered in  [Wu et al., 2021], while the only difference is to replace the Gram matrix in  [Wu et al., 2021] with kernel matrix.

**2.** It is true that the SGD algorithm considered in this paper is different from the epoch-wise SGD in [Wu et al., 2021], but it seems that the proof technique will not be that different as claimed by the authors. One difference I can see is that when using SGD in steps, one can focus on the expected components of the iterates along different directions, while for SGD in epochs, one can directly calculate the components rather than taking expectations. The authors may need to clearly clarify the difference in the Introduction.

**3.** Can you provide some intuition regarding the case when the gram matrix is not diagonal dominant? Does the result still hold or can the proof technique still work?

**4.** Experiments in figure 2 are basically the same as those in [Wu et al., 2021], could you clarify some differences between them? Otherwise, these experiments may not be necessary since they do not provide any new message.

**5.** Regarding Assumption 1, it would be better to provide some example kernel functions and data models right after this assumption rather than deferring them to Appendix.

==========================

Thanks for your response. I suggest you revise your paper accordingly at least to highlight the difference and novelty compared to [Wu et al., 2021] .  Besides, it could be better if you can consider more challenging settings (e.g., the gram matrix is not that diagonal dominant), which could be closer to the practical cases.

**Summary Of The Paper:**

This paper studies the directional bias of SGD in kernel regression. In particular, this paper shows that when using moderate or small step size, GD converges along the direction corresponding to the smallest eigenvalue of the covariance matrix. In contrast, when provided with a moderate initial learning rate with annealing, SGD converges along the direction corresponding to the largest eigenvalue. Consequently, the authors show that such directional bias of SGD can result in an estimator that is closer to the ground truth, which further leads to better generalization.

**Summary Of The Review:**

Given my concern about the novelty and contribution of this paper, I do not recommend acceptance.

---

> ### Author Response · Authors · 2021-11-20
> **Reply to Reviewer 2**
>
> We thank the reviewers for the comments.
> 1. Admittedly, the formulation is similar.
>
> 2.  We admit the difference is not so big, but our result is still useful since their proof does not automatically holds for step-wise SGD, while step-wise SGD is used in practice and our theorem can explain the interesting phenomenon of directional bias in this case.
>
> 3. When the gram matrix is not diagonal dominant, the proof no longer holds. However, as seen from our experiments, the directional bias may still exists in practice.
>
> 4. The experiments in Figure 2 use ResNet, while [Wu et al, 2021] use a CNN without the residual block structure. Considering that the ResNet is popular in ML community, our experiment provide new and useful message for this case. And this experiment is necessary as we explained in Remark 13 why this experiment is related to the theorems in our paper.
>
> 5. In current version, we include one example of kernel function right after Assumption 1, but still leave other examples in Appendix due to page limit.

---

### Official Review · Reviewer_RGP1 · 2021-11-03

**Correctness:** 3
**Technical Novelty And Significance:** 3
**Empirical Novelty And Significance:** 2
**Recommendation:** 5
**Confidence:** 4

**Main Review:**

The paper is interesting and deals with a very important issue: generalization in simple problems, and the effects of training with SGD compared to GD. Focusing on simple tasks is very powerful and definitely we need more insights in this direction. However, I think the paper has a few drawbacks and deserves a few months of more work. Here are some concerns/questions for the authors. I will read the review, discuss with the authors and potentially upgrade my score.

1) The main concern I have in the link to generalization and the link to properties in the *training* landscape. Theorem 8 is trivial and is well known, however the link to generalization definitely does not follow: A is the training matrix, and GD converges exactly to w^* – the problem is that the w^* in test loss is different! In general, even though i might be missing something here, I invite the authors to tone the claim down: "explaining generalization" is the the holy grail of ML, such cold claims are somewhat unacceptable in published research.

2) Theorem 4 and 6 are nice and novel, however how do they compare? I have read Remark 7 but I do not feel like this is good enough: for SGD you use 2 different stepsizes and you also have a particular choice dependent on the C_i constants. My question is simple: lets take eta = 0.01 for both GD and SGD, fixed during learning. does the remark still hold? why? Please, show this experimentally (yes you can, and its a different setting compared to figure 1)!

3) Going exactly to experiments: why is there no experiment on generalization for kernel regression? your paper is about that! I would be surprised if you find that SGD generalizes better than GD in a fair setting (i.e. you tune the methods, and you use the same tricks for both).

4) Second experimental concern: why in your FMNIST experiment you do not select the same stepsizes ranges for GD and SGD? You do not show GD performance with the "moderate stepsize" you have in SGD

5) Suggestions/ typos
- explain better directional bias in the intro
- typo in sentence "we prove that a two-stage SGD has b_t converges in the direction of the largest eigenvector of K"
- again typo  "that GD hasbt converges" before thm 6


**Summary Of The Paper:**

The paper studies the trajectory of GD and SGD for kernel regression, and the connection to generalization. The paper shows that, under some stepsize choices, SGD has a directional bias towards sharp eigendirections of the kernel matrix. The authors link this feature to generalization using the properties of quadratics.

**Summary Of The Review:**

The topic of the paper is very nice, but I have some concerns and hope to have a nice conversation with the authors!

---

> ### Author Response · Authors · 2021-11-20
> **Reply to Reviewer 1**
>
> We thank the reviewer for the careful comments, and we reply the comments as following:
> 1. We agree with the reviewer that generally the $w^*$ in test loss and empirical risk are different -- unless we have infinitely many data for training then $w^*$ in empirical risk converges to the minimizer of the test loss. The reason why we can directly link the estimation error and generalization error here is that we consider the special noiseless case, then the minimizer of the test loss, denote $w^*$ -- which is the ground truth -- will also be the unique minimizer for the empirical risk.
>
> 2. We have compared SGD and GD with the same step size in our experiment in Figure 2, and empirically the result in Theorem 4 and 6 are still true. Admittedly, such comparison is not done in Figure 1 for our previous version, so in current version we add such comparison.
>
> 3. We add comparison for generalization performance in Figure 1 in current version, and the generalization error of SGD is smaller than GD, though the training loss of GD is smaller than SGD.
>
> 4. In the experiment for FMNIST, the stepsizes range for GD and SGD are the same, and we have experiment for GD with moderate stepsize.
>
> 5. We add one more reference for explaining the direction bias in the intro, and we correct the typo in current version.

---

> > ### Comment · Reviewer_RGP1 · 2021-11-21
> > **Thanks for the response**
> >
> > Thanks for the response!
> >
> > 1) Then you are not talking about generalization loss, right? I am not sure then this work targets what people usually call "directional bias".
> >
> > 2) Thanks for the interesting experiments. However, in the appendix I read that you did not compare SGD and GD with the same exact stepsizes: "GD is of magnitude n/2 times that of SGD". This is indeed in your theorem, but makes the comparison weak: the stepsizes you use for SGD can be boosted (dynamics is not noisy at all) – in this case, probably the RQ would converge to the same value. Indeed,  looking at figure 1(a), it seems that the RQ has not converged!
> > Also, why is the test accuracy so bad?
> >
> > I will keep my score for now, lets see after the internal discussion!

---

> > > ### Author Response · Authors · 2021-11-23
> > > **Thanks for the comments**
> > >
> > > We thank the reviewer for the comments.
> > >
> > > 1. We are not sure what is the 'directional bias' that the reviewer think of, is it not covered in the references that we surveyed in our work? We wonder if the reviewer can provide some references about the directional bias that we are not aware of?
> > >
> > > 2. We will try to do more experiment in the future to address these issue.
> > >
> > > Thanks!

---

### Decision · Program_Chairs · 2022-01-20

**Decision:**

Reject

**Comment:**

This is an interesting paper, aiming to separate the generalization properties of SGD and GD.  Unfortunately, the reviewers had many significant concerns, primarily on the topic of the relationship to prior work by Wu et al. (which has a similar setting and similar proof techniques), but also regarding presentation and interpretation of results in general.  As such, I recommend the authors continue with this line of valuable work, aiming in particular to further separate it from existing results.